



# Stochastic Wake Modeling Based on POD Analysis

David Bastine[1], Lukas Vollmer[2], Matthias Wächter[1], and Joachim Peinke[1]

[1]AG TWiSt, Institute of Physics, ForWind, University of Oldenburg, Küpkersweg 70, 26129 Oldenburg
[2]AG Energy Meteorology, Institute of Physics, ForWind, University of Oldenburg, Küpkersweg 70, 26129 Oldenburg

*Correspondence to:* David Bastine (david.bastine@posteo.de) or Matthias Wächter (matthias.waechter@uni-oldenburg.de)

**Abstract.** In this work, large eddy simulation data is analyzed to investigate a new stochastic modeling approach for the wake of a wind turbine. The data is generated by the LES model PALM combined with an actuator disk with rotation representing the turbine. After applying a proper orthogonal decomposition (POD), three different stochastic models for the weighting coefficients of the POD modes are deduced resulting in three wake models. Their performance is investigated mainly on the
basis of aeroelastic simulations of a wind turbine in the wake. Three different load cases and their statistic characteristics are compared for the original LES, truncated PODs and the stochastic wake models including different numbers of POD modes. It is shown that approximately six POD modes are enough to capture the load dynamics on large temporal scales. Modeling the weighting coefficients as independent stochastic processes leads to similar load characteristics as in the case of the truncated POD. To complete this simplified wake description, we show evidence that the small-scale dynamics can be grasped by adding
to our model a homogeneous turbulent field. In this way we present a procedure, how to derive stochastic wake models from costly CFD calculations or elaborated experimental investigations. These numerically efficient models provide the added value of possible long-term studies. Depending on the aspects of interest, different minimalized models may be obtained.

## 1 Introduction

More and more wind turbines are organized in large wind farms containing up to hundreds of turbines. Consequently, a large
number of these turbines frequently operates in the wake of other turbines. The reduced wind speed and enhanced turbulence in the wake flow leads to power losses (Barthelmie et al., 2007, 2010) and increased fatigue loadings (Frandsen, 2005). Therefore, modeling of wakes plays a key role in the design process of wind turbines and entire wind farms (Barthelmie et al., 2009; Chowdhury et al., 2012; Gonzalez et al., 2014; Schmidt, 2014) as well as in the emerging field of wind farm control (Corten and Schaak, 2003; Fleming et al., 2014).

Wake modeling through fully resolved simulations based on the Navier-Stokes equations is computationally very expensive due to the many scales relevant to turbulent flows (Frisch, 1995). The most detailed dynamical simulations which can be computed in reasonable times are large eddy simulations (LES) (Pope, 2000) combined with simplified turbine models, such as actuator disk or actuator line models (Mikkelsen, 2003; Calaf et al., 2010; Lu and Porte-Agel, 2011; Wu and Porte-Agel, 2011; Porté-Agel et al., 2011; Goit and Meyers, 2014; VerHulst and Meneveau, 2014; Witha et al., 2014a). Even though these
simulations have proven to be an efficient tool for the investigation of specific research questions, LES are still too time-





consuming for most practical applications. In particular, long-time studies cannot be performed with such demanding CFD tools. Therefore, much simpler wake models are needed which strongly reduce the computational costs.

A variety of wake models exist which only describe the steady mean velocity deficit in the wake flow. They range from simple kinematic models (Jensen, 1983; Frandsen et al., 2006) over approximated versions of the Reynolds averaged Navier-Stokes (RANS) equations to combinations of the full RANS equations with simplified turbine models (Schmidt, 2014). While these steady models can be used for estimations of the power output, the loads acting on turbines in the wake cannot be calculated from the mean velocity deficit since they strongly depend on the dynamics of the wake flow.

In industry applications the wake dynamics are often taken into account by modeling the additional turbulence intensity caused by the presence of the wake (Quarton and Ainslie, 1990; Frandsen, 2005). Such a single quantity, however, can obviously not describe all relevant dynamical features of a wake flow. One possible approach to a more precise description is given by the dynamic wake meandering model (DWM) (Larsen et al., 2007b, 2008; Madsen et al., 2010; Keck et al., 2015). It consists of three major elements, namely a model for a steady velocity deficit, a model describing large-scale movement of the wake caused by large atmospheric structures and a model for the added turbulence caused by the rotor.

The DWM shows some promising results (e.g. Larsen et al., 2013), but it still remains an open question which features of the wake flow have to be taken into account. In particular, the influence and interplay of different large-scale effects has not yet been understood. For example, laboratory (Singh et al., 2014) and field measurements (Bastine et al., 2015a) indicate that the turbine modulates the atmospheric flow on a wide range of scales, even on scales up to five or more rotor diameters (Singh et al., 2014).

Another dynamic approach, also followed in this work, is to analyze and model wind turbine wakes using modal decompositions (Andersen et al., 2012, 2013; Bastine et al., 2014, 2015b; Hamilton et al., 2015, 2016; Iungo et al., 2015; Sarmast et al., 2014; VerHulst and Meneveau, 2014) which describe the velocity field as a linear superposition of spatial modes with time-dependent weighting coefficients. It has been shown that a few spatial modes can already capture important features of the wake flow (Andersen et al., 2012, 2013; Bastine et al., 2014, 2015b; Hamilton et al., 2015; Iungo et al., 2015). In the case of a large eddy simulation of an infinite row of turbines, Andersen et al. (2012) found that a few modes stemming from the proper orthogonal decomposition (POD) already yield a good description of the velocity field on large spatial scales. For PIV-Data obtained in a wind turbine boundary layer array, Hamilton et al. (2015) have shown that a few modes can approximately reproduce the spatial dependence of the Reynolds stress tensor. In Bastine et al. (2015b), dynamical features of quantities relevant for a sequential turbine in the wake, such as the energy flux through a disk, could be captured well with only three modes.

Most of the works on decompositions mentioned above only deal with reduced descriptions of the wake while approaches to model the temporal evolution have rarely been investigated. The temporal dynamics of reduced order systems stemming from modal decompositions are completely described by the weighting coefficients of the selected modes (Berkooz et al., 1993). For relatively simple fluid flows, a system of corresponding differential equations can be obtained by projecting the Navier-Stokes Equation on selected POD modes (Berkooz et al., 1993; Cordier et al., 2013). For the wind turbine wake, this projection is difficult to handle due to the complex interaction of the flow and the wind turbine. Furthermore, the description and inclusion of a turbulent atmospheric inflow is a very challenging task. An alternative approach, investigated by Iungo et al.



(2015), is to linearize the time evolution leading to the dynamic mode decomposition (Schmid, 2010; Jovanović et al., 2014). In this framework, the relevant weighting coefficients are all periodic. Iungo et al. (2015) extended this approach by embedding the reduced system within a Calman-filter making data-driven modeling possible. For an LES of an infinite row of turbines, dominant frequency peaks have been found for power spectral densities of the weighting coefficients of POD modes. Andersen

(2014) and Andersen et al. (2012) tried to model the weighting coefficients by simply using only these periodic parts.

In far and intermediate wake regions of a single wake with a turbulent inflow, dominating periodic oscillations are not necessarily present, as indicated by e.g. Singh et al. (2014) and Iungo et al. (2013). Since turbulent flows such as wind turbine wakes show only statistically reproducible results (Frisch, 1995), this work suggests a new approach modeling the weighting coefficients of a POD as a stochastic process yielding a stochastic wake description. This idea is investigated through the

analysis of large eddy simulations (LES) of an actuator disk with rotation (Witha et al., 2014b) in a turbulent atmospheric boundary layer (ABL). The obtained POD modes are combined with simple stochastic models for the weighting coefficients. Since we are mainly interested in the impact of the wake flow on sequential turbines, aeroelastic simulations of a wind turbine in the wake are performed. Original LES, truncated PODs and stochastic models are used as inflows and the results are compared for different numbers of modes included. Furthermore, we investigate the problem of missing turbulent kinetic energy in the

modeled wake flow, which is a principle shortcoming of reduced order models based on modal decompositions. We illustrate that it is principally possible to capture the small-scale properties of the flow by adding a homogeneous turbulent field to the wake structure modeled by the POD-based approach.

The article is structured as follows. The LES used in this work is described in Sect. 2. Subsequently in Sect. 3, we introduce the methods necessary to obtain the stochastic wake models from the LES data. Furthermore, we illustrate how the performance

of the different wake descriptions is investigated based on aeroelastic simulations. The analysis of the LES data begins in Sect. 4 with a standard POD analysis followed by an investigation of the performance of truncated PODs depending on the number of included modes. Subsequently in Sect. 5, we deduce the three different stochastic wake models based on the LES data and compare their performance to truncated PODs and the original LES. To mimic the small-scale wake turbulence we add an additional homogeneous turbulent field to one of the stochastic wake models in Sect. 6 and analyze the performance of this

extended model. Conclusions from our results are drawn in Sect. 7.

## 2   LES Simulations

The large eddy simulations have been performed using the **PA**rallelized **L**ES **M**odel PALM (Raasch and Schroter, 2001; Maronga et al., 2015) which has been extensively used for the simulation of the atmospheric boundary layer for the last 15 years. PALM solves the non-hydrostatic, incompressible Navier-Stokes Equations under the Boussinesq Approximation using

central differences on a uniformly spaced Cartesian staggered grid. For the time integration a third-order Runge-Kutta scheme and for the advection terms a fifth-order Wicker-Skamarock scheme is used. Subgrid-scale turbulence is filtered implicitly and is parametrized by a modified Smagorinsky approach following Deardorff (1980).



Recently, PALM has been combined with simplified wind turbine models for the investigation of wind turbine wakes and the simulation of entire wind farms (Witha et al., 2014a, b; Dörenkämper et al., 2015; Vollmer et al., 2016). Here, an enhanced actuator disk model with rotation (ADM-R) is used which provides close results to an actuator line model in the far wake while being much less computationally expensive (Wang, 2012; Witha et al., 2014b). The ADM-R parameters are set to model

the NREL 5 MW research turbine (Jonkman et al., 2009) with a hub height of $z_h = 90\,\mathrm{m}$ and a rotor diameter of D = 126 m. Adaptation of the rotor speed to the fluctuating wind speed is ensured by a variable-speed generator-torque controller (Jonkman et al., 2009).

The flow field for the simulation with the wind turbine model is established in a pre-run with cyclic boundary conditions that is initialized with a laminar wind profile and is run for 18 hours of simulation time until it has reached a quasi-steady state. The

development of turbulence is initiated by random perturbations at the beginning of the pre-run. The simulations are run with a roughness length of $z_0 = 2 \cdot 10^{-3}\,\mathrm{m}$, representative for a medium rough sea surface, a neutral potential temperature profile capped by an inversion at 500 m and a Coriolis parameter for $\phi_{lat} = 54°\,\mathrm{N}$. A uniform grid size of 5 m is chosen with 1024 grid points in along-stream and 512 grid points in cross-stream direction. The time step used for the integration is $dt = 0.3$ s and the analyzed data includes 23500 snapshots corresponding to 7050 s.

A turbulent recycling method (Maronga et al., 2015) is used for the simulation with the wind turbine to enable a simulation of a single turbine instead of simulating an infinite row. For this purpose, the domain size is doubled along the along-stream axis. The recycling surface is placed at the domain length of the precursor run. Undisturbed outflow at the downstream boundary is ensured by a radiation boundary condition.

The mean velocity field far upstream the turbine is shown in Fig. 1a and the corresponding profile in Fig. 2a. At hub height

the average velocity is approx. $8\,\frac{\mathrm{m}}{\mathrm{s}}$, with a turbulence intensity of approx. $5\%$. In the following sections, we analyze the stream-wise component $u$ of the wake flow in the yz-plane 3.5 D away from the turbine. Snapshots of this plane are shown in Fig. 3 revealing a variety of shapes of the wake structure. The mean velocity at hub height is approx. $4\,\frac{\mathrm{m}}{\mathrm{s}}$, as can be seen in the mean velocity field shown in Fig. 1b and the corresponding profile in Fig. 2b. The turbulence intensity at hub height is approx. $16\%$. In the outer region of the wake a strongly increased variance turbulent of the field is observed (Fig. 1c and Fig. 2c).

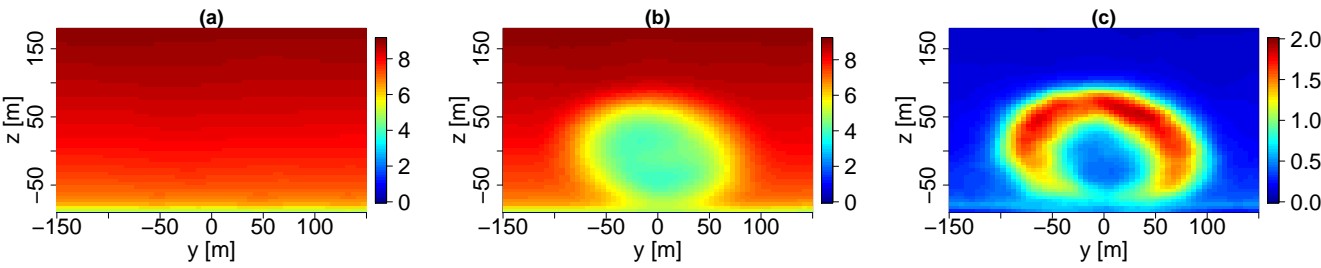

**Figure 1.** Statistics of the stream-wise velocity $u$ [ms$^{-1}$]: (a) Mean field far upstream the turbine (b) Mean field 3.5 D away from the turbine (c) Variance [m$^2$s$^{-2}$] 3.5 D away from the turbine.





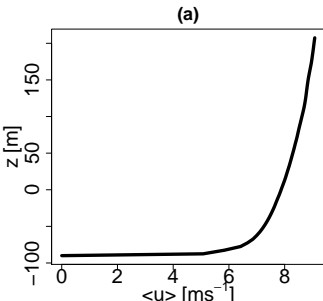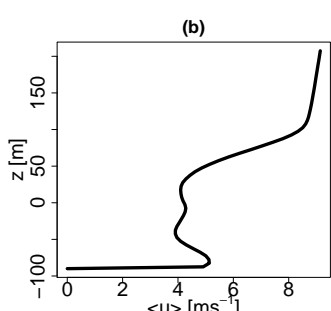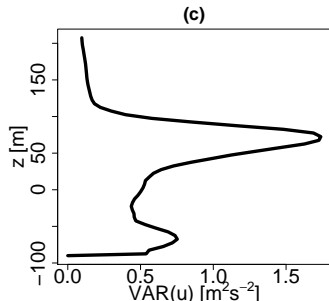

**Figure 2.** Different profiles at $y = 0$ m corresponding to the figures in Fig. 1: (a) Mean field far upstream the turbine (b) Mean field 3.5 D away from the turbine (c) Variance 3.5 D away from the turbine.

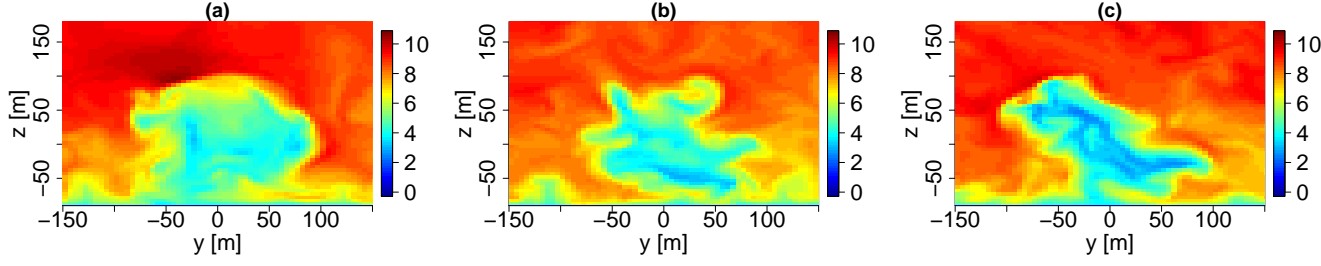

**Figure 3.** Snapshots of the LES showing the the stream-wise velocity $u$ [ms$^{-1}$] in a yz-plane 3.5 D away from the turbine. (a) $t = 990$ s (b) $t = 1185$ s (c) $t = 1230$ s .

## 3   Methods

In this work, a stochastic wake modeling approach is developed to capture the dynamics of a wind turbine wake in a new manner. For this purpose, spatial modes stemming from a POD are combined with weighting coefficients modeled as stochastic processes. The necessary steps and methods involved in building such wake models based on LES Data are presented in Sect.

5  3.1-3.4. Several assumptions and simplifications are made in this process which in the end have to be justified by a satisfying performance of the model. In Sect. 3.5, we explain how model and original LES flow will be compared to draw conclusions on the performance of the model. Finally, the connections between load dynamics and a dynamic inflow, which is described by a modal decomposition, are discussed.

### 3.1   Preprocessing

10  Before the POD is applied to the data, the velocity field is preprocessed similarly as in Bastine et al. (2015b) to focus the analysis on the wake structure. The preprocessing is illustrated in Fig. 4. First, we subtract the mean field far upstream the





turbine (Fig. 1a) from the wake flow (Fig. 4a). The velocity deficit obtained after changing the sign of the field is shown in Fig. 4b. Second, we extract the deficit by using a (temporally local) relative threshold. This means that we set all values smaller than $40\%$ of the current deficit maximum to zero. This extraction is followed by a dilation procedure to keep the neighboring regions which are higher than the threshold. The resulting extracted deficit is shown in Fig. 4c.

It should be noted that the stochastic modeling approach, presented in the following, does not principally rely on the chosen preprocessing procedure. Instead of choosing a threshold, we also performed an analysis where the analyzed region is confined to a fixed circular region around the wake center. Similar results have been obtained in this case. The threshold procedure is chosen to be consistent with our former work presented in Bastine et al. (2015b) where it lead to better results concerning the selection of POD modes.

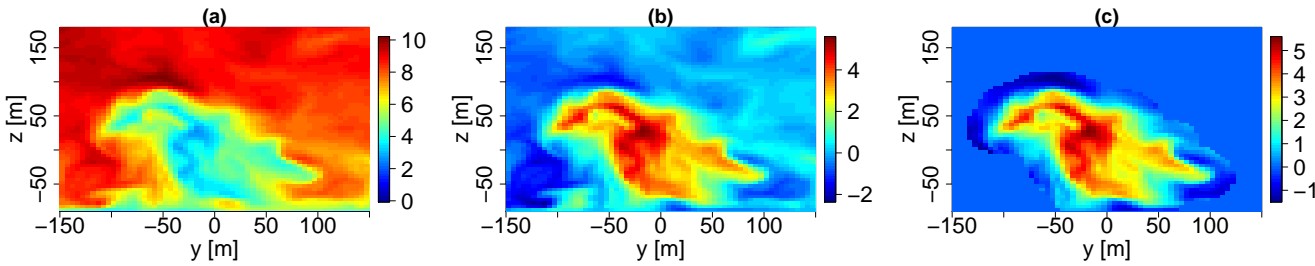

**Figure 4.** Preprocessing of the velocity field $u\,[\mathrm{ms}^{-1}]$ : (a) Instant Snapshot $t = 31.8$ s (b) Velocity deficit (c) Extracted deficit.

## 3.2   Proper Orthogonal Decomposition (POD)

A decomposition of the velocity field $u(y,z,t)$ into spatial modes with time-dependent weighting coefficients can be written as:

$$u(y,z,t) = \langle u(y,z,t)\rangle_t + \sum_{j=1}^{\infty} a_j(t)\phi_j(y,z) , \qquad (1)$$

where $\langle ...\rangle_t$ denotes averaging over time. In case of the POD, the $\phi_j(y,z)$ are called POD modes which can be defined as the
eigenfunctions of the covariance operator solving:

$$\int dy' dz' \, \langle u'(y,z,t)u'(y',z',t)\rangle_t \, \phi_j(y',z') = \lambda_j \phi_j(y,z) \qquad (2)$$

with $u' = u - \langle u\rangle_t$. The covariance operator is a compact self-adjoint operator yielding a countable number of real eigenvalues which are usually ordered as $\lambda_1 > \lambda_2 > ...$ . The corresponding orthogonal POD modes $\phi_j(y,z)$ can also be chosen as real-valued functions with corresponding weighting coefficients $a_j(t)$ obtained through the projection:

$$a_j(t) = (\phi_j|u') := \int dy' dz' \, \phi_j(y',z') u'(y,z,t) . \qquad (3)$$





It should be noted that in case of performing the POD on simulation data on a spatial grid, as done here, Eq. (2) is commonly approximated by the eigenvalue problem of the discretized covariance matrix $C_{ij} = \langle u_i^{'}(t) u_j^{'}(t) \rangle_t$. where $u(y,z)$ is substituted by $u_k$ as the value of $u$ at the k-th grid point.

When aiming for a reduced description of the velocity field it is common practice to truncate the POD after $N$ modes yielding:

$$u^{(N)}(y,z,t) := \langle u(y,z,t) \rangle_t + \sum_{j=1}^{N} a_j(t) \phi_j(y,z)) \approx u(y,z,t). \tag{4}$$

For such approximations of the field $u$, POD modes solving Eq. (2) are the optimal modes with respect to the turbulent kinetic energy since they minimize the mean squared error given by

$$\langle \| u'(y,z,t) - \sum_{j=1}^{N} a_j(t) \phi_j(y,z) \|_2^2 \rangle_t , \text{ with } a_j(t) = (\phi_j | u') . \tag{5}$$

Another important property of the POD is that the temporal behavior of the weighting coefficients is uncorrelated:

$$\langle a_i(t) a_j(t) \rangle_t = \lambda_i \delta_{ij} \tag{6}$$

which simply comes from the fact that the covariance operator is diagonal in the basis of its own eigenvectors.

Using Eq. (4) the POD offers a systematic, and in an energetic sense, optimal way to reduce the complexity of the velocity field. The number of modes $N$ needed to obtain a useful approximation of the flow is often much lower than the number of grid points. This leads to a strong dimensional reduction of the system. For our case of a wind turbine wake in the $yz$-plane, a typical grid point number is in the order of $10^3 - 10^4$ with approximately 40 modes needed to grasp $80\%$ of the turbulent kinetic energy, as for example discussed in Andersen (2014) and Bastine et al. (2015b). In this work, we often aim for a reduction to a system with 10 or less modes which will be referred to as using only "a few" modes in the following.

### 3.3 Temporal Stochastic Modeling

Although the POD can lead to a reduced description of the field, no modeling of the temporal dynamics is involved. A truncated POD as in Eq. (4) can simply be viewed as a special kind of spatially filtered field. Nonetheless, the POD naturally suggests an approach for the description of the temporal dynamics since all dynamical information lies in the weighting coefficients $(a_j(t))_{i=1}^{N}$ of the $N$ selected modes. Therefore, in this paper we aim for an efficient way to model the time-dependence of these weighting coefficients. Our ansatz is to describe these $N$ weighting coefficients as a stochastic system. As we expect a high degree of complexity from a turbulent wake, this is a promising approach. All the missing information such as left out weighting coefficients, velocity data on other grid points etc. is lumped into statistical fluctuations. In this work, we simplify the $N$-dimensional stochastic system by assuming statistical independence of all $a_j(t)$ yielding $N$ one-dimensional systems. Even though this assumption is inspired by Eq. (6), it obviously leads to a significant approximation since the nonlinear coupling of different scales in the fluid dynamical equations is neglected. It therefore has to be justified by a satisfying performance of the deduced model.





In the following, $\tilde{a}_j(t)$ denotes the stochastic process (or a corresponding realization) which models the i-th weighting coefficient. The symbol $a_j(t)$ denotes the time series stemming from the projections of the original LES on the POD modes (Eq. (3)). Inserting $\tilde{a}_j(t)$ instead of $a_j(t)$ into a truncated POD (Eq. (4)) leads to a stochastic wake model given by

$$\tilde{u}^{(N)}(y,z,t) = \langle u(y,z,t)\rangle_t + \sum_{j=1}^{N} \tilde{a}_j(t)\phi_j(y,z) . \tag{7}$$

Next, we introduce three different stochastic models for the $\tilde{a}_j(t)$ yielding three different stochastic wake models. We start with an almost trivial model for $\tilde{a}_j(t)$ given by independent Gaussian random numbers with the mean $\tilde{\mu}_j$ and the variance $\tilde{\sigma}_j^2$. We choose $\tilde{\mu}_i = \langle a_j(t)\rangle_t = 0$. For the only free parameter $\tilde{\sigma}_j$, we choose the estimated standard deviation of the original $a_j(t)$ yielding

$$\tilde{\sigma}_j = \sqrt{\langle a_j(t)^2\rangle_t} . \tag{8}$$

The wake model resulting from inserting $\tilde{a}_j$ into Eq. (7) is called the *uncorrelated model* in the following. Since realizations of $\tilde{a}_j(t)$ are discontinuous time series, the *uncorrelated model* also leads to velocity fields which are discontinuous in time.

As a slightly more complex model, we now use an Ornstein-Uhlenbeck process (e.g. Risken, 1984; Gardiner et al., 1985) which is defined by

$$\dot{\tilde{a}}_j(t) = -k_j\tilde{a}_j(t) + \gamma_j\xi(t) , \text{ with } \langle\xi(t+\tau)\xi(t)\rangle = \delta(t-\tau) , \ k_j > 0 \tag{9}$$

where $\xi(t)$ is Gaussian white noise and $k_j$ and $\gamma_j$ are the parameters of the model. In this work, the integration of Eq. (9) is done simply by using the analytically known two-point probability density function (pdf) (see. e.g. Gardiner et al. (1985)). The correlation function of $\tilde{a}_j$ is given by:

$$c_j(\tau) := \langle\tilde{a}_j(t+\tau)\tilde{a}_j(t)\rangle_t = \frac{\gamma_j^2}{2k_j}e^{-k_j|\tau|} \tag{10}$$

with the variance $\langle\tilde{a}_j(t)^2\rangle = \frac{\gamma_j^2}{2k_j}$ and the correlation time

$$\tau_{a_j} := \int_0^\infty d\tau\, c_j(\tau) = \frac{1}{k_j} , \tag{11}$$

also called integral time scale. These relations can be used to estimate parameters $k_j$ and $\gamma_j$ by:

$$k_j = \frac{1}{\tau_{a_j}} \tag{12}$$

$$\gamma_j = 2\langle a_j^2(t)\rangle_t\tau_{a_j} , \tag{13}$$

where $\tau_{a_j}$ will be roughly estimated through $c(\tau_{a_j}) = \frac{1}{e}$. This way the variance and the integral time scale of the original $a_j(t)$

are approximately reproduced. However, this does not mean that the second order two-point statistics of the original $a_j(t)$





such as the auto-correlation function or the power spectral density (PSD) are also matched well. The stochastic wake model corresponding to the $\tilde{a}_j(t)$ described as Ornstein-Uhlenbeck processes will be referred to as the *OU-based model*. Even though time series of the Ornstein-Uhlenbeck process are continuous they are still non-differentiable due to the fast fluctuations of the white noise. The *OU-based model* therefore also yields non-differentiable velocity fields.

The third model is based on a parametrized power spectral density resulting in a better description of the original PSD and other two-point statistics of the $a_j(t)$, as will be discussed further in Sect. 5.1. The PSD of the $\tilde{a}_j$ are given by:

$$S^{(j)}(f) = \frac{S_0^{(j)}}{1 + \left(\frac{f}{f_{\frac{1}{2}}^{(j)}}\right)^{\alpha^{(j)}}} \tag{14}$$

with the parameters $S_0$ (power at zero) and $f_{\frac{1}{2}}$ (frequency at $\frac{S_0}{2}$). For higher frequencies, $S$ behaves like a power law with an exponent given by parameter $\alpha$. The fitting procedure to obtain these parameters for all included weighting coefficients is

shortly described in Sec 5.1. The phases $\phi_j(f)$ of the $\tilde{a}_j$ are modeled as independent uniformly distributed random variables between 0 and $2\pi$ yielding the Fourier transform $\hat{\tilde{a}}(f) = S(f; S_0, f_{\frac{1}{2}}, \alpha)e^{i\phi(f)}$ where the indices $(j)$ are discarded for reasons of clarity. An inverse Fourier transform yields the differentiable time series $\tilde{a}_j(t)$. The corresponding wake model, obtained from inserting the $\tilde{a}_j(t)$ into Eq. (7) will be called *spectral model* in the following.

Our stochastic ansatz is not principally confined to these relatively simple models for the weighting coefficients. More

complex stochastic processes, as described in e.g. Friedrich et al. (2011) and Kantz and Schreiber (2004), could be chosen which reproduce nonlinear moments of the $a_j$ or allow for a coupling between the different weighting coefficients. However, a lot of data is needed to obtain reliable estimates of the parameters corresponding to such more complex models. Furthermore, a higher number of parameters might make it more difficult to build a practically applicable wake model. The three stochastic wake models introduced here are investigated in Sect. 5.

**3.4   A Spectral Surrogate in Three Dimensions**

One common feature of truncated modal decompositions used in this work is that a certain fraction of missing turbulent kinetic energy typically on smaller scales. In order to take these small-scale dynamics into account, in Sect. 6 an additional turbulent field is combined with the *spectral model* introduced in the former section. The additional turbulent field is a three dimensional spectral surrogate of the original LES field confined to a central spatial region (see Sect. 6 for more details). Let this confined

field be called $u_c(y, z, t)$. The surrogate is obtained by keeping the absolute values of the Fourier transform $|\hat{u}_c(k_y, k_z, f)|$ while substituting the phases $\phi(k_y, k_z, f)$ with uniformly distributed random numbers between 0 and $2\pi$. An inverse Fourier transformation yields the surrogate field. This way, the PSD and all the second order correlations of the field are conserved. Spatial inhomogeneities on the other hand are lost since the spectral surrogate is a stationary and spatially homogeneous field. With this surrogate field we investigate the general possibility of using a homogeneous turbulent field.





### 3.5 Aeroelastic Simulations and Model Verification

Our main motivation for modeling wakes is to draw conclusions on the impact on other wind turbines. Therefore, we use aeroelastic simulations to model a wind turbine in the wake flow. For these simulations, we use the open source software FAST, developed by NREL, and its embedded subroutines of the aerodyn code (Laino, 2005) which are based on the blade

element momentum (BEM) theory (e.g. Burton et al., 2011). As inflow, we use the original LES data, truncated PODs and the output of the stochastic wake models. In this paper, we use three of the multiple loads calculated by FAST , namely the rotor torque $T$, the rotor thrust $F_t$ and the tower base yaw moment in z-direction $t_z$. The $v$ and $w$ components ($y-$ and $z-$direction) of the inflow, which we do not model here, are set to zero. For the original simulation, setting $v$ and $w$ to zero lead to almost no differences in the output of FAST at least for the aspects we investigated here. It is clear that there are cases where the $v, w$-

components become important. For such aspects our modeling procedure can be extended to these components in a similar manner.

To draw conclusions on the performance of the stochastic wake models we compare their calculated loads with the loads for truncated PODs and the original LES. Since we model the wake in a stochastic manner, statistical properties of the loads are compared.The distribution of energy over different time scales is investigated by an estimation of the PSD which is obtained via

averaging the absolute squared Fourier spectrum over 20 windows using a cosine shaped weighting function. Furthermore, the entire energies in the load signals is compared through their variances. Subsequently, we use the algorithm given by Nieslony (2010) to estimate rainflow counting histograms (ASTM, 1994) since they are commonly used to draw conclusions on the life time of wind turbines. The rainflow counting histograms will simply be called RFCs in the following. Based on the RFC calculations, damage equivalent loads (DELs) are estimated yielding the constant load amplitude necessary to cause the same

cumulative damage as the investigated load time series (for a specific number of cycles $N_{eq}$). The damage equivalent load is given by

$$\text{DEL} = \left( \frac{\sum\limits_i n_i S_i^m}{N_{eq}} \right)^{\frac{1}{m}}. \tag{15}$$

where $n_i$ is the number of cycles with amplitude $S_i$ and $m$ is the so called Wöhler exponent (e.g. Burton et al., 2011) Here, we choose $m = 10$ which is a typical value for materials used for WEC rotors. Since we compare the DELs resulting from reduced

wake descriptions to the DEL resulting from the original LES ( $\text{DEL}_0$), we only consider normalized DELs given by

$$\frac{\text{DEL}}{\text{DEL}_0} = \left( \frac{\sum\limits_i n_i S_i^m}{\sum\limits_i n_{0,i} S_{0,i}^m} \right)^{\frac{1}{m}}. \tag{16}$$

Additionally, we analyze the variance $\langle u'(y,z)^2 \rangle_t$ of the different wake descriptions since it is one of the most significant features of a turbulent wake flow . For the original LES, it has already been shown in Fig. 1a. $\langle u'(y,z)^2 \rangle_t$ is also the first diagonal entry of the Reynolds stress tensor and a measure for the local average turbulent kinetic energy in the stream-wise

component. In the rest of this work, it will thus be referred to as the local TKE. In contrast to the characteristics of loads on turbines in the wake flow, $\langle u'(y,z)^2 \rangle_t$ is a property of the flow itself.



In summary, we will compare in this work different wake descriptions with the original LES based on the measures introduced above. Truncated PODs are analyzed in Sect. 4.2, the three different stochastic wake models in Sect. 5.2 and an extended stochastic wake model with added turbulence in Sect. 6.2. As a direct property of the flow, the local TKE of the different descriptions is considered. The impact on a turbine in the wake is investigated based on three different loads: the rotor torque, the
rotor thrust and the tower base yaw moment. These loads are compared for the different wake descriptions by analyzing: time series, PSDs, RFCs, variances of the time series and DELs.

### 3.6 Time-dependence of Loads

In this section we shortly discuss the dynamical behavior of loads and its relation to a dynamic inflow which is described by a modal decomposition, such as $u^{(N)}(y,z,t)$ in Eq. (4) or $\tilde{u}^{(N)}(y,z,t)$ in Eq. (7). This discussion will enable us to gain a deeper
understanding of the results presented in the next sections 4-6.

The temporal evolution of loads strongly depends on the inflow a wind turbine experiences. The time-dependence of this inflow is mainly determined by two mechanisms. First, the flow structures in the rotor plane change in time due to the hydrodynamics of the flow field, particularly due to the advection through the rotor plane. This time-dependence of the field $u^{(N)}(y,z,t)$ is completely described by the time-dependence of the $(a_j(t))_{i=1}^N$ and obviously leads to time-dependent loads.
Second, the turbine experiences a changing velocity field since the rotor blades move through an inhomogeneous velocity field and its flow structures. This rotation often causes partially periodic behavior of the load time series leading to peaks in the corresponding PSDs which are multiples of the average rotational frequency $\langle f_{\mathrm{rot}} \rangle_t$. It is an interesting question, whether for specific velocity fields one of the described mechanisms plays a more important role than the other for the load dynamics.

Based on this discussion, we conclude that the statistical properties of a load time series are determined by both, the statistical
properties of the $a_j(t)$ and the spatial characteristics of the field which are mainly determined by the POD modes $\phi_j$. However, these two contributions cannot be considered completely separately. Obviously, the spatial characteristics of the field are also influenced by the $a_j(t)$ since they represent the amplitudes of the normalized spatial structures $\phi_j(y,z)$. Particularly, the energy $\langle a_j^2(t) \rangle_t = \lambda_j$, as a measure for the energetic relevance of $\phi_j$, plays an important role. Additionally, the dynamic characteristics of the $a_j(t)$ determine the influence of the spatial structures on the loads. For example, a stationary field $u$ with constant $a_j(t)$
clearly leads to purely periodic behavior of the loads yielding sharp spectral peaks in the PSDs of loads. On the other hand, weighting coefficients changing on time scales $\tau_{a_i} << \frac{1}{3\langle f_{\mathrm{rot}} \rangle_t}$ obviously lead to less pronounced peaks since the inflow has strongly changed during one third of a rotation. Therefore, the dynamic characteristics of the $a_j(t)$ influence both, the explicit time-dependence of the $u(y,z,t)$ and the time-dependence of a turbine's inflow caused by the moving blades.

For the influence of the moving blades, the rotational speed of the rotor plays an important role. Together with the charac-
teristic length scales of coherent structures it determines relevant time scales for the load dynamics. Similarly, the advection speed interacts with the spatial characteristics in $x$-direction yielding the temporal characteristic of the $a_j(t)$. The advection speed and the rotational speed often differ strongly. A typical tip speed ratio of current turbines is around 7 (Burton et al., 2011). Therefore, even statistically homogeneous isotropic structures can be responsible for energy contributions in the load signals on completely different time scales.



## 4    Truncated PODs

In this section, we apply the POD, introduced in Sect. 3.2, to the LES data described in Sect. 2. The obtained POD modes and corresponding eigenvalues are briefly described in Sect. 4.1. In Sect. 4.2, we investigate how many POD modes are necessary to grasp important aspects of the wake flow. For this purpose, truncated PODs (Eq. (4)) including different numbers of modes

are analyzed via the different aspects introduced in Sect. 3.5.

### 4.1    POD Modes and Eigenvalues

We solve the eigenvalue problem in Eq. (2) for the preprocessed velocity field, described in Sect. 3.1. This yields the eigenvalues shown in Fig. 5a and the POD modes shown in Fig. 6. While 10 modes capture approx. $50\%$ of the turbulent kinetic energy of the preprocessed field, almost 100 modes are needed to capture $90\%$ (Fig. 5b). This reflects that the energy of the flow is

distributed over a wide range of scales, which is a typical property of turbulent flows. There is slight tendency from larger to smaller structures with increasing mode number which is even more pronounced when mode numbers $j > 10$ are investigated. The modes seem more complex than the modes obtained in Bastine et al. (2015b). This is likely to be caused by a stronger interaction with the ground since the hub height in Bastine et al. (2015b) was 160 m, in contrast to 90 m in the simulation here. This interaction also breaks the rotational symmetry of the wake deficit yielding statistics which are not invariant under

rotations, as already visible in the variance of the field in Fig. 1c. The POD modes also reveal this non-symmetric behavior of the wake. Statistically axisymmetric fields lead to modes which are axisymmetric themselves or form statistically axisymmetric subspaces combined with other modes which have the same eigenvalue (Berkooz et al., 1993). As discussed in Bastine et al. (2015b), mode 1 is related to the horizontal large-scale motion of the wake. The non-axisymmetry of the wake is revealed e.g. by the fact that we do not find a similar mode representing the motion in another direction.

The eigenvalues presented in Fig. 5a equal the variance of the weighting coefficients due to Eq. (6). Thus, the energy in the fluctuations of the $a_j(t)$ also decreases with mode number. A simple quantity characterizing the dynamical behavior of the $a_j(t)$ is the integral time scale $\tau_c$ (Eq. (11)) shown in Fig. 5c. The integral time scale also decreases with mode number corresponding to faster fluctuations with increasing $j$. In the spirit of frozen turbulence this could also be understood as a corresponding decrease of the length scales in $x$-direction.





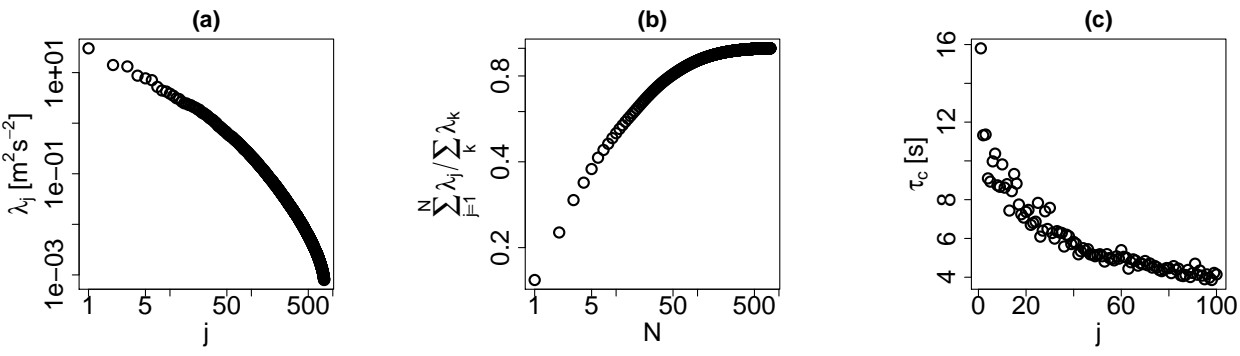

**Figure 5.** (a) POD eigenvalues (b) Normalized cumulative spectrum of the POD representing the percentage of captured turbulent kinetic energy (c) Integral time scale of the weighting coefficients versus mode number.

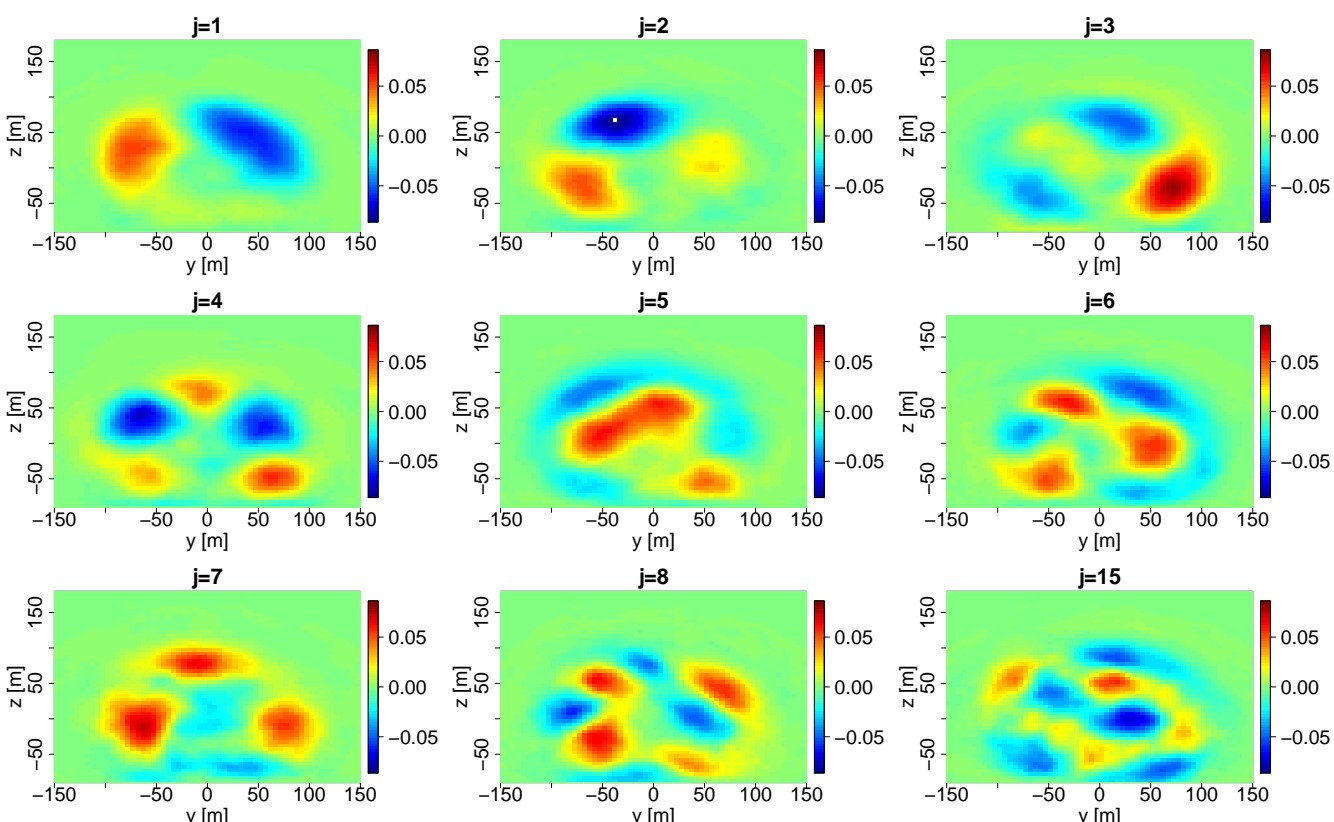

**Figure 6.** POD modes $\phi_j(y, z)$.



## 4.2 Performance of Truncated PODs

We now use only a finite number $N$ of the obtained POD modes yielding truncated PODs (Eq. (4)) as an approximate description of the original wake flow . The corresponding weighting coefficients are given by Eq. (3), i.e. their temporal evolution is directly calculated from the LES data. Thus, no temporal modeling takes place. Including different numbers of modes $N$,

the truncated PODs are now compared to the original LES based on the different aspects introduced in Sect. 3.5. A brief presentation of the results is followed by a more detailed discussion.

As Hamilton et al. (2015), we find that the spatial dependence of the local TKE $\langle u'(y,z)^2 \rangle_t$ of the wake flow can already be captured by a few modes (Fig. 7), in our case approx. four modes. It can also be seen that many more modes are necessary for capturing the magnitude of $\langle u'(y,z)^2 \rangle_t$.

As described in Sect. 3.5, the truncated PODs are also used as inflows for aeroelastic simulations. The simulated load time series of torque $T$, thrust $F_t$ and tower base yaw moment $t_z$ are shown in Fig. 8. For a truncated POD with 6 modes, they roughly follow the corresponding signals of the original LES. More precisely, less than 10 modes are necessary to capture the load dynamics on large temporal scales, as illustrated through the PSDs of the loads shown in Fig. 9. All PSDs show a similar behavior with a flat region for low frequencies followed by a decay and peaks which are approximate multiples of the average

rotational frequency $\langle f_{\text{rot}} \rangle_t \approx 0.12$ Hz. The low frequency region is relatively well matched for all loads, with the weakest performance for the tower base yaw moment $t_z$. The small temporal scales, however, cannot be grasped and thus the truncated POD misses a large part of the energy in the load signals which can also be seen by looking at the variances in Fig. 10a. For 10 modes, we get less than $60\%$ of the variances of $F_t$ and $t_z$. Less than $90\%$ are captured for $40$ modes. For the torque $T$, already 20 modes will give approx. $100\%$ .

The RFCs of the different loads are shown in Fig. 11. While for $T$ the RFCs approximately match the original loads when including 6 modes or more, strong deviations can be seen for the thrust $F_t$ and the tower base yaw moment $t_z$. Particularly, the occurrence of large amplitudes cannot be captured even when including 20 modes. Consequently, a lot of modes are also needed to capture the DELs for $F_t$ and $t_z$ (Fig. 10b). 40 modes for example yield $90\%$ of the DEL for the original LES. As for the variance, less modes are needed for the torque $T$ reaching more than $85\%$ with 6 modes.

The interpretation of the results above is started by giving a possible explanation for capturing the spatial structure of the local TKE with only a few modes. We suspect that the strongly enhanced turbulence in the outer region of the wake is mainly caused by the fact that this region is sometimes covered by the wake and sometimes experiences free flow. A few modes can describe this effect qualitatively, since they capture the large-scale motion and the dynamics of the coarse shape of the wake

(Bastine et al., 2015b). However, the magnitude of turbulence cannot be grasped since the small-scale turbulence is missing in the wake deficit of a truncated decomposition.

The capturing of large-scale dynamics of the load time series can be understood well by following the ideas presented in Sect. 3.6. The large temporal scales of the loads are strongly related to large temporal scales of the $a_j(t)$ and to the rotor blades moving through large spatial structures. Larger structures are mostly found in lower order POD modes which also





correspond to $a_j(t)$ with relatively long integral time scales (Sect. 4.1). Therefore, the discarded higher order modes contain only smaller structures and correspond to faster fluctuating $a_j(t)$. Consequently, a few modes can capture the load behavior on large temporal scales but fail to capture the small-scale behavior.

The missing small-scale fluctuations in the load signals are directly related to the weak performance of the truncated PODs to reproduce variance, RFCs and DELs. Hence for the RFCs, our results also indicate that large rainflow amplitudes do not correspond to the dynamics on a specific frequency region but to the dynamics on various time scales.

The generally simpler performance for the torque might be related to the large moment of inertia of the rotor causing the higher frequencies, which are poorly captured, to play a less important role. This can be seen by the relatively low frequency peaks in Fig. 9a and the relatively smooth time series in Fig. 8a. Hence, the time scales relevant to a specific load strongly influence the number of modes necessary for a satisfying description of this load.

The higher number of modes needed for a good large-scale description of $t_z$ might be caused by the importance of the wake position for this load. This has been discussed in Bastine et al. (2015b) for a simplified measure related to $t_z$. It has been shown that the horizontal motion can be captured qualitatively with only one specific mode but that many modes are needed to describe the magnitude of the motion.

Particularly for the rotor torque, the variance and damage equivalent loads also reveal that the inclusion of some modes lead to a strong improvement while others yield almost no effect. Therefore, a further dimensional reduction might be possible by only selecting modes relevant to the specific load of interest. This has also been discussed in Bastine et al. (2015b); Saranya-soontorn (2005, 2006).

In summary, we showed that relevant aspects of wake flow could be well described by only a few modes. Particularly, the spatial dependence of the local TKE and the large temporal scales of the different load dynamics could be well captured. This is an important first step for a wake description of reduced order and we try to reproduce these results in Sect. 5 using the stochastic wake models from Sect. 3.3. RFCs, variances and DELs of the loads are not well described with only a few modes due to the missing energy in the small scales of the wake. This aspect is further investigated in Sect. 6.





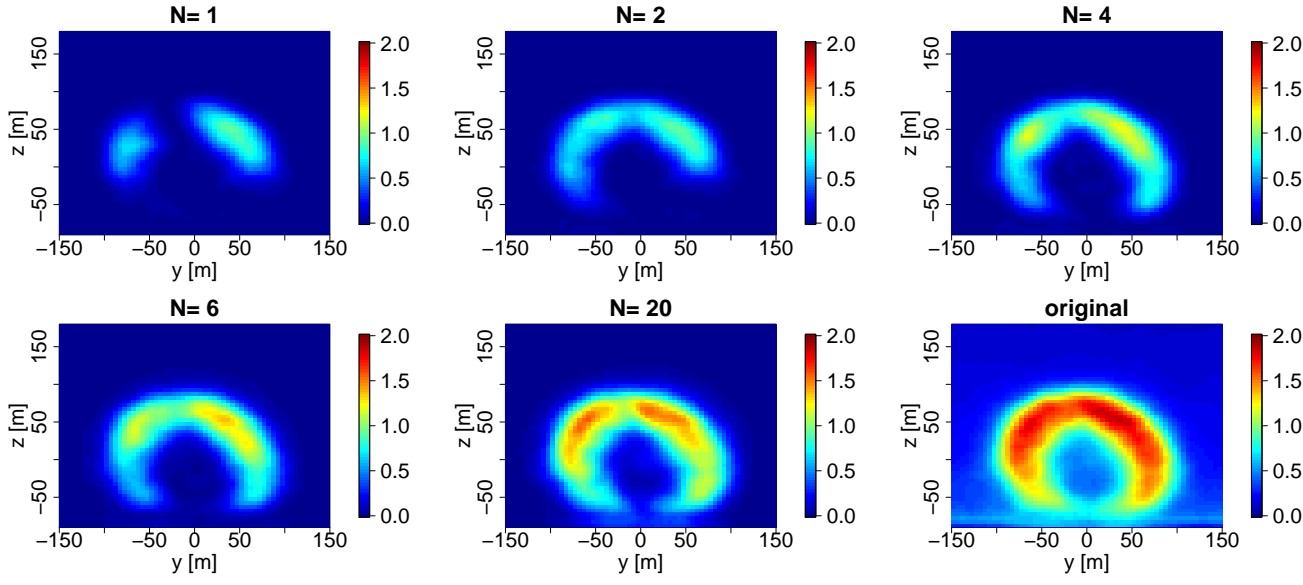

**Figure 7.** Local TKE $\langle u'(y,z)^2 \rangle_t$ [m$^2$s$^{-2}$] for original LES and truncated PODs including different numbers of modes $N$.

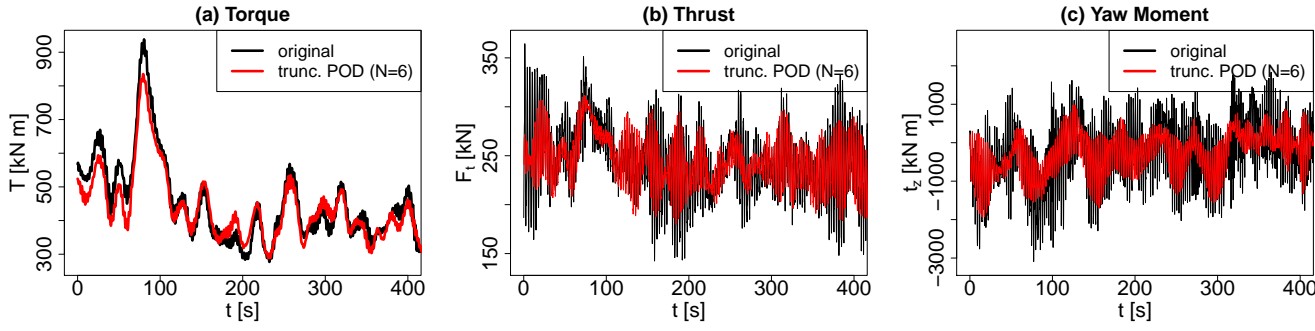

**Figure 8.** Time series of the different loads for original LES and a truncated POD including $N = 6$ modes.




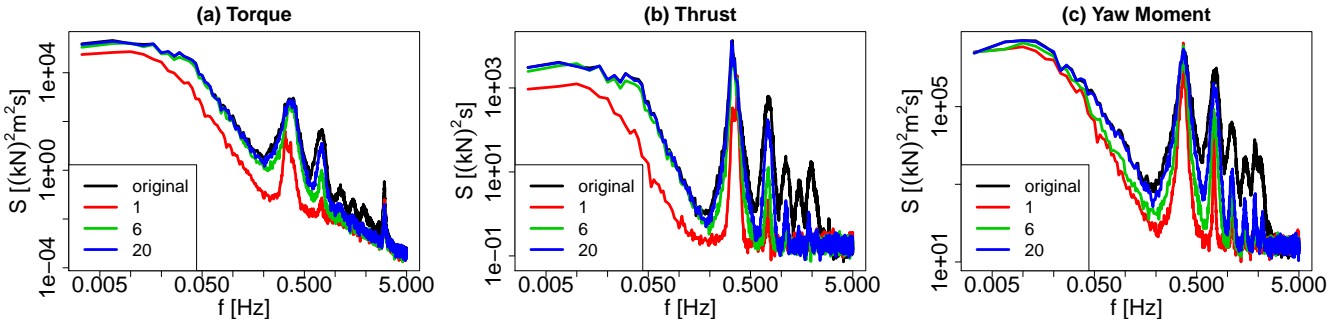

**Figure 9.** PSDs of the different loads for original LES and truncated PODs including different numbers of modes $N$.

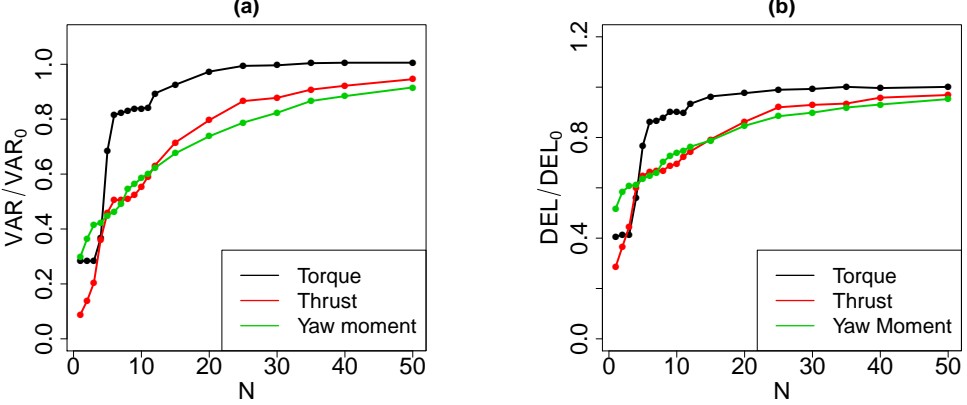

**Figure 10.** (a) Variance and (b) damage equivalent loads (DELs) versus the number of modes included in the truncated POD. Both are normalized by the values of the original LES.




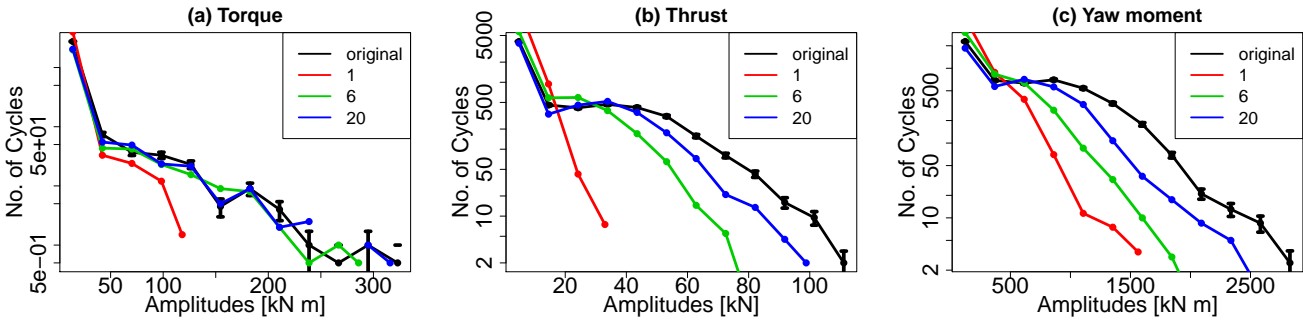

**Figure 11.** Rainflow counting histograms (RFCs) of different load time series for original LES and truncated PODs including different numbers of modes. To get an impression of the estimation error, the standard error shown for the original LES is estimated by $\frac{\sqrt{n_i}}{2}$ where $n_i$ is the number of half-cycles in bin $i$.

## 5 Stochastic Wake Models

In this section, three different stochastic wake models , as introduced in Sect. 3.3, are deduced from the LES data and their performance is investigated. In Sect. 5.1, the weighting coefficients $a_j(t)$ which were used in the truncated POD are analyzed further to obtain the model parameters for the corresponding stochastic descriptions $\tilde{a}_j(t)$. Using the estimated parameters and resulting time series $\tilde{a}_j(t)$ in the decompositions leads to the three stochastic wake models which are compared with the truncated POD and the original LES in Sect. 5.2. This comparison is done on the basis of the local TKE and the aeroelastic quantities introduced in Sect. 3.5.

### 5.1 Modeling the Weighting Coefficients $a_j(t)$

We now deduce the model parameters for the *uncorrelated model*, the *OU-based* model and the *spectral model*. Subsequently, we shortly investigate the ability of these models to capture statistical properties of the original weighting coefficients.

As discussed in Sect. 4.1, the estimated variance and integral time scales of the $a_j(t)$ decrease with mode number (Fig. 5a and Fig. 5c) which corresponds to weaker but faster fluctuations. For the *uncorrelated model*, the $\tilde{a}_j$ are completely determined by the variance $\langle a_j(t)^2 \rangle_t$ due to Eq. (8). The parameters $k_j$ and $\gamma_j$ for the *OU-based model* can be deduced from variance and integral time scales using Eqs. (13).

For the *spectral model*, the PSDs of the $a_j$ have to be estimated. They show a qualitatively similar behavior for all $j$ starting with a flat region for low frequencies followed by an approximate power law behavior (Fig .12a). This form motivates the parametrization of the PSDs given by Eq. (14). The parameters $S_0$, $\alpha$ and $f_{\frac{1}{2}}$ are estimated using least squares in the log-log framework fitting $\log(S)(\log(f); S_0, f_{\frac{1}{2}}, \alpha)$. While this yields satisfying estimates of $\alpha$ and $f_{\frac{1}{2}}$, $S_0$ is systematically underestimated due to the logarithmic function. We circumvent this problem by choosing $S_0$ to be the value which yields the estimated variance of the $a_j(t)$: $\mathrm{VAR}[\tilde{a}_j(t; S_0)] = \langle a_j(t)^2 \rangle_t$. An example fit is shown in Fig. 12b. The estimated parameters for varying



$j$ are shown in Fig. 13. $S_0$ shows a fast decrease with mode number. Therefore, the decrease of the variance of the $a_j$ with $j$ is mainly related to a decrease of fluctuations on large temporal scales. The power law exponent $\alpha$ shows a negative trend yielding a slight increase for the energy related to higher frequencies. The increase of $f_{\frac{1}{2}}$ is related to a decrease of the integral time scale $\tau_c$ of the $a_j$.

Next, we investigate the ability of the three different models to reproduce properties of the original $a_j(t)$. Realizations of the different $\tilde{a}_j(t)$ for $j = 2$ are shown in Fig. 14. The time series for the *spectral model* shows a qualitatively similar behavior as the $a_2$ obtained from the LES. For the Ornstein-Uhlenbeck process a similar integral time scale can be suspected but faster fluctuations play an important role yielding a non-differentiable time series. The *uncorrelated model* simply yields random

10    numbers with the correct variance. In the following, variance, integral time scales and PSDs of these time series are compared.

The estimated variance $\langle a_j(t)^2 \rangle_t$ is approximately matched by all three models (Fig. 15a) since all three models fulfill $\mathrm{VAR}[\tilde{a}_j] = \langle \tilde{a}_j(t)^2 \rangle_t$ by definition. The integral time scale can be approximately captured by the *OU-based* and the *spectral model*, as shown in Fig. 14. For the *uncorrelated model*, the integral time scales are exactly zero by definition.

The fitting procedure for the *spectral model* yields a good description of the PSDs up to $0.2$ Hz, as illustrated by the nor-

15    malized PSDs in Fig. 12c. For $j = 2$ estimated PSDs and the analytically calculated PSDs corresponding to the estimated parameters are shown in Fig. 15c and Fig. 12b, respectively. These figures show that the PSD for the *OU-based model* strongly differs from the original PSD. Thus, even though the integral time scale and variance are well matched by the *OU-based model*, the distribution of energy over the different scales is different. The uncorrelated model yields a trivial PSD also failing to reproduce the PSD of $a_2$. Similar results as for $j = 2$ are found for other mode numbers.

Overall, the *spectral model* obviously does the best job describing the statistical properties of the original $a_j$ yielding smooth differentiable time series with similar second order two-point statistics. This has been expected since it is the most complex model using three model parameters to fit the original data.



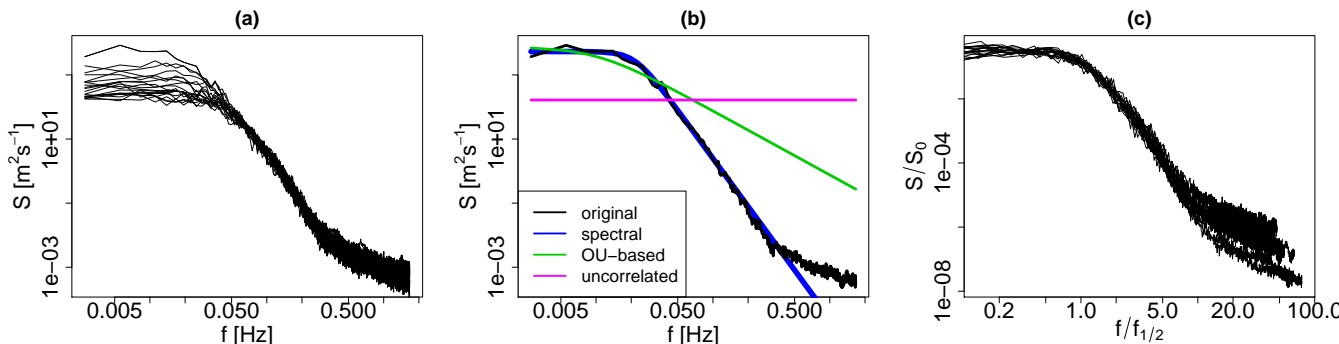

**Figure 12.** (a) PSDs of the weighting coefficients $a_j(t)$ for $j = 1, 2, ..., 50$ (b) PSD for $a_2(t)$ and the corresponding fit for the *spectral model*. Additionally, analytical PSDs for *uncorrelated* and *OU-based model* are shown corresponding to the estimated model parameters of these models. (c) Normalized PSDs of $a_j(t)$ for $j = 1, 2, ..., 50$. The estimated parameters for $S_0^{(j)}$ and $f_{\frac{1}{2}}^{(j)}$ are used to normalize $S$ and $f$, respectively.

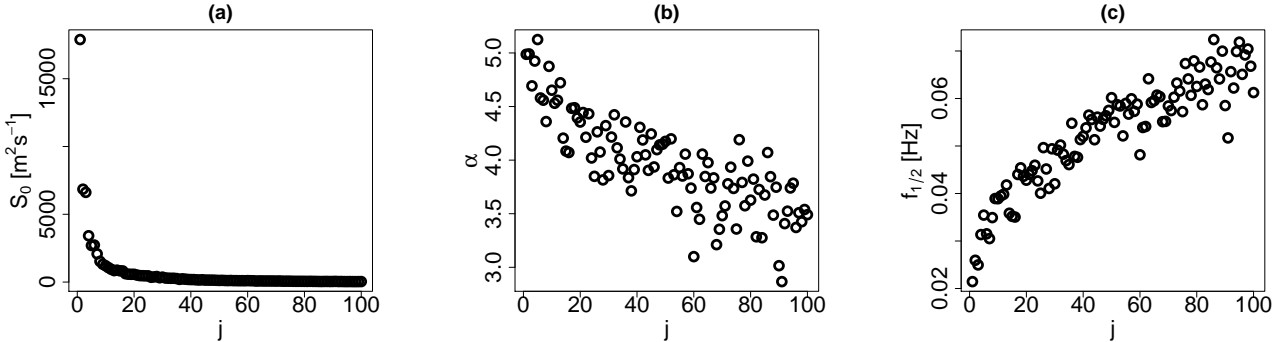

**Figure 13.** Estimated parameters for the *spectral model* versus the mode number $j$.





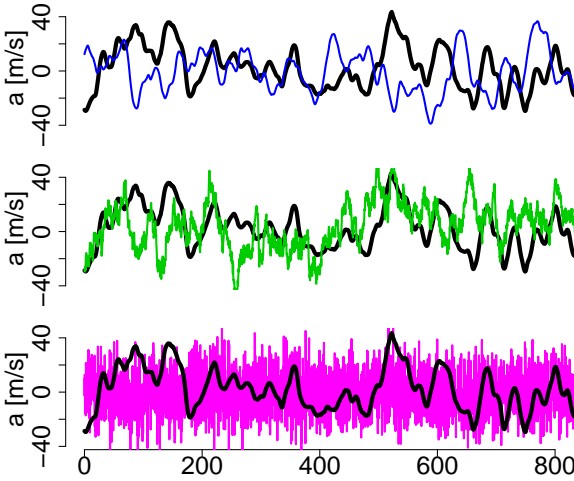

**Figure 14.** Time series of $a_2(t)$ (black) and the different stochastic models $\tilde{a}_2(t)$. From top to bottom: *spectral model* (blue), *OU-based model* (green) and *uncorrelated* model (magenta).

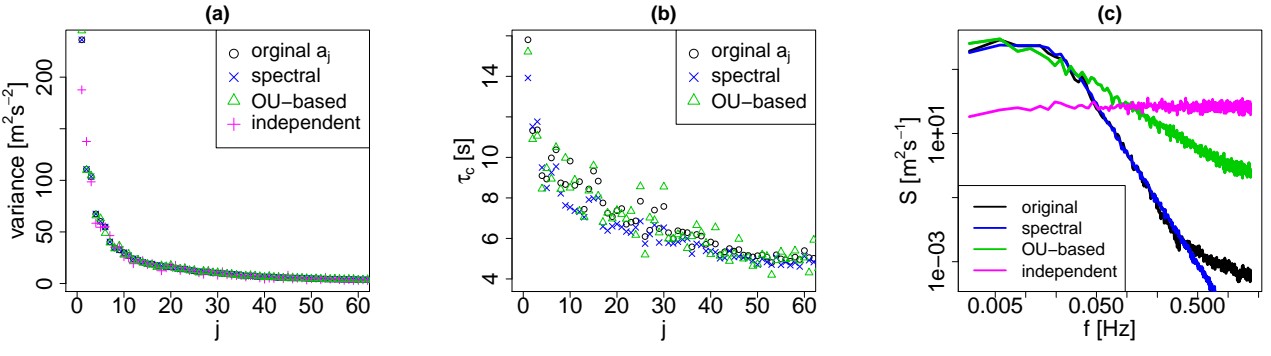

**Figure 15.** Properties of the weighting coefficients $a_j(t)$ and the stochastic models $\tilde{a}_j(t)$: (a) Variance versus mode number $j$ (b) Integral time scales versus mode number. The results for the *uncorrelated model* are not shown since the integral time scale is zero for all $j$. (c) Estimated PSDs of $a_2(t)$ and realizations of the different stochastic models $\tilde{a}_2(t)$.

## 5.2 Performance of the Stochastic Wake Models

Based on the model parameters estimated above, we can now generate random time series $\tilde{a}_j(t)$ and insert them into Eq. (7) yielding realizations of the corresponding stochastic wake models, introduced in Sect. 3.3. In this section, we investigate the performance of these models with respect to the aspects specified in Sect. 3.5. A brief presentation of the results is followed
5 by a more detailed discussion.



The $\tilde{a}_j(t)$, used for the stochastic wake models, aim for capturing statistical properties of the $a_j(t)$ used in the truncated POD. Therefore, a stochastic wake model $\tilde{u}^{(N)}$ including $N$ modes performs well if it yields similar results as the corresponding truncated POD $u^{(N)}$. Hence, we compare the outcome for the truncated POD and the different stochastic wake models for a fixed number $N$. Here, $N = 6$ is presented since this number already lead to promising results for the truncated POD, as

discussed in Sect. 4.2. Later in this section, the performance of the *spectral model* will also be investigated for varying $N$.

The local TKE $\langle \tilde{u}'(y,z)^2 \rangle_t$ looks very similar for the truncated POD and all three stochastic wake models (Fig. 16) showing only minor quantitative differences which could be caused by statistical fluctuations.

Sections of the time series for the loads from aeroelastic simulations are shown in Fig. 17. The *uncorrelated model* yields fast fluctuating loads in contrast to the loads of the truncated POD which change on larger time scales. For the *OU-based-* and

*spectral model*, the time series resemble the loads of the truncated POD but drawing further conclusions from a single short time window is difficult.

The PSDs of the time series, shown in Fig. 18, reveal different behavior for the different models. While the *spectral model* coincides with the truncated POD, the *OU-based model* shows significant differences. Particularly, it has a smaller slope around $0.1$ Hz and underestimates the energy in the low frequency regime. On the other hand, the width and height of the peak

at $3 \cdot \langle f_{\mathrm{rot}} \rangle_t$ are relatively well matched. As expected, the *uncorrelated model* fails almost completely to reconstruct the PSD.

Examining the RFCs in Fig. 19, the *uncorrelated model* also shows a strongly different behavior than the truncated POD . The RFCs for *spectral* and *OU-based model* approximately coincide with the RFC of the truncated POD. Differences for the *OU-based model* can only be suspected and do not seem to be significant.

For the *spectral model*, we also investigate the behavior for different numbers of included modes with respect to the variance

and DELs, as shown in Fig. 20. For less than ten modes, truncated POD and stochastic model show similar results. For higher mode numbers, variance and DELs of the rotor torque $T$ appear to be underestimated by the model while a slight overestimation is present for the thrust $F_t$. However, also larger estimation errors are present for higher mode numbers. The shown errors are estimated as the standard deviation for an ensemble of $10$ realizations of the *spectral model*, calculated for $N = 1, 4, 5, 6, 10, 30$. It should be noted that the statistical estimates for the truncated PODs also have errors which are more difficult to estimate but

are expected to be of the same order of magnitude as for the *spectral model*.

We start the interpretation of the results with an explanation for the similar local TKE found above. This result is caused by the fact that $\langle \tilde{u}'(y,z)^2 \rangle_t$ depends only on the variance of the weighting coefficients due to:

$$\langle \tilde{u}'(y,z)^2 \rangle_t \tag{17}$$

$$= \langle \sum_{i=1,j=1}^{N} \tilde{a}_i(t) \tilde{a}_j(t) \phi_i(y,z) \phi_j(y,z) \rangle_t = \sum_{i=1,j=1}^{N} \langle \tilde{a}_i(t) \tilde{a}_j(t) \rangle_t \phi_i(y,z) \phi_j(y,z) \tag{18}$$

$$= \sum_{i=1,j=1}^{N} \langle \tilde{a}_i(t)^2 \rangle_t \delta_{ij} \phi_i(y,z) \phi_j(y,z) = \sum_{i=1}^{N} \langle \tilde{a}_i(t)^2 \rangle_t \phi_i(y,z)^2 \tag{19}$$

Since the variance of the $a_j$ is approximately matched by the $\tilde{a}_j$ of all three stochastic wake models, the local TKE is matched as well.





It is not surprising that the *spectral model* which is based on the PSD of the $a_j$ does the best job capturing the PSDs of the loads and that the *uncorrelated model* containing no information about two-point correlations fails almost completely. However, a more detailed understanding, particularly for the *OU-based model*, might be possible when following the ideas from Sect. 3.6. Based on this discussion, we suspect that the *OU-based model* performs well around $3\langle f_{\mathrm{rot}}\rangle_t$ because this

frequency regime is dominated by the movement of the blades through the velocity field and that for this movement mainly the persistence time of the POD structures is important, rather than the exact temporal two-point statistics. This persistence time is given by correlation time of the $a_j(t)$ which is matched well by the *OU-based model* (Sect. 5.1).

For the RFCs, matching the variance and correlation time of the $a_j(t)$ might also be sufficient since the *OU-based* model performed relatively well. Possibly, the frequency region around $3\langle f_{\mathrm{rot}}\rangle_t$ and thus the rotation of the blades through the wake

field plays a dominant role for the appearing cycles in the time series. In other words, the spatial characteristics of the flow in the $yz$-plane might be more important than the temporal characteristics or the characteristics in the $x$-direction.

The satisfying performance of the *spectral model* for $N \leq 10$, as illustrated by variance and DELs, shows that our results are not confined to the case of $N = 6$. It is difficult to identify the reason for the possibly weaker performance when including more modes. One possible explanation might be that the parametrization of the PSD or the fitting procedure might be less good

for higher mode numbers. However, preliminary experiments with spectral surrogates of the $a_j$, which match the PSD exactly, indicate similar trends. Another reason could be that the assumption of independence of the $a_j$ becomes problematic when including many modes.

In summary, we showed that modeling the weighting coefficients as independent stochastic processes can lead to similar

statistical results as obtained when using the original weighting coefficients used in the truncated POD. Similar local TKE as well as PSDs, RFCs of the three different loads could be obtained for the *spectral model* which approximately captures the PSD of the original weighting coefficients. For the RFCs, we might even need less complex stochastic processes since simply capturing the integral time scales and variances of the $a_j(t)$ lead to promising results. Completely neglecting two-point correlations, however, can only reproduce a similar local TKE structure as illustrated by the results of the *uncorrelated*

*model*. It should be noted that the stochastic wake models still have the same shortcomings as the truncated POD, namely the missing energy for small-scale dynamics when including only a few modes. Furthermore, they might also perform weaker when including many modes. An alternative approach circumventing the inclusion of a large number of modes is investigated in the next section.



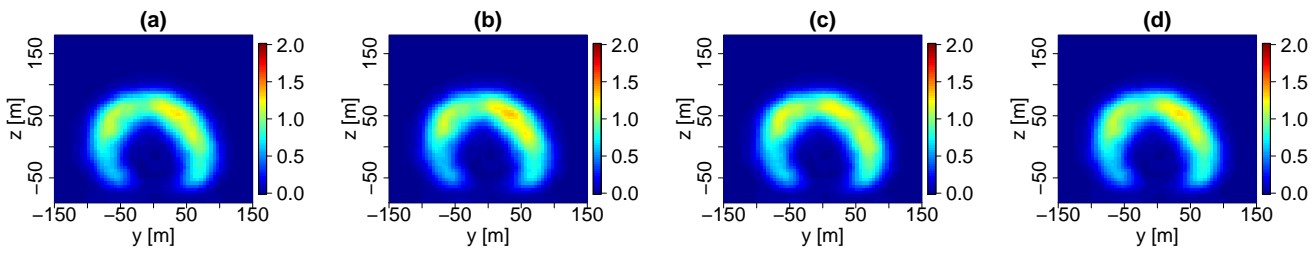

**Figure 16.** Local TKE $\langle u'(y,z)^2 \rangle_t$ $[\mathrm{m^2 s^{-2}}]$ using $N = 6$ POD modes for truncated POD and the different stochastic wake models: (a) truncated POD (b) *spectral model* (c) *OU-based model* (d) *uncorrelated model*.

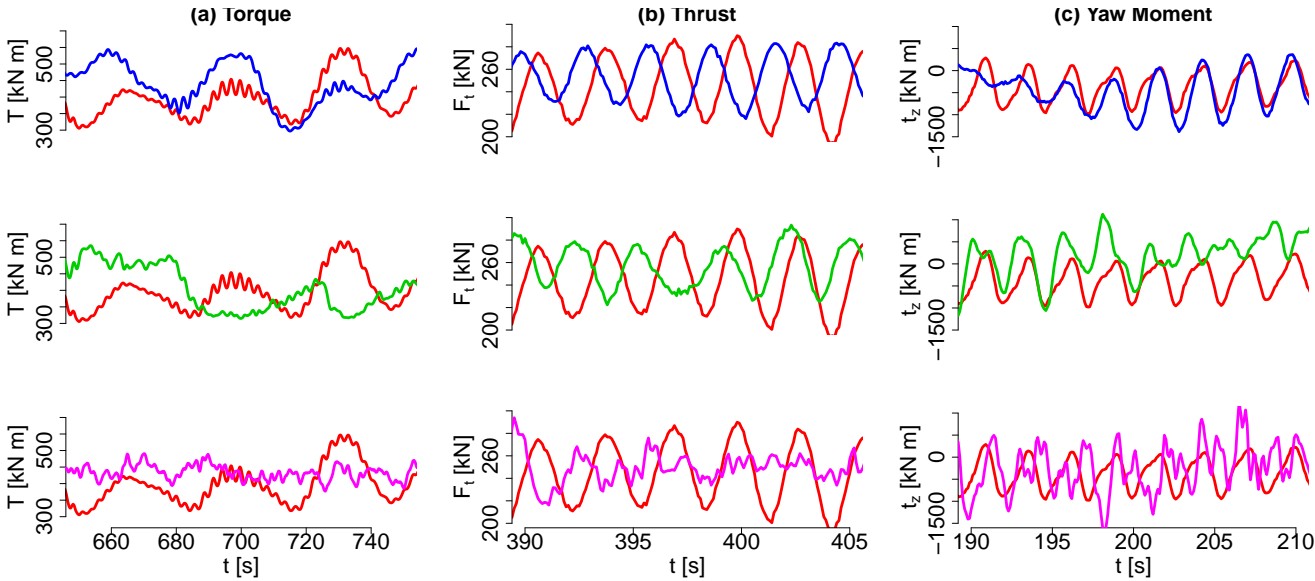

**Figure 17.** Time series of the different loads for the truncated POD (red) compared to *spectral model* (blue), *OU-based model* (blue), *uncorrelated model* (magenta).





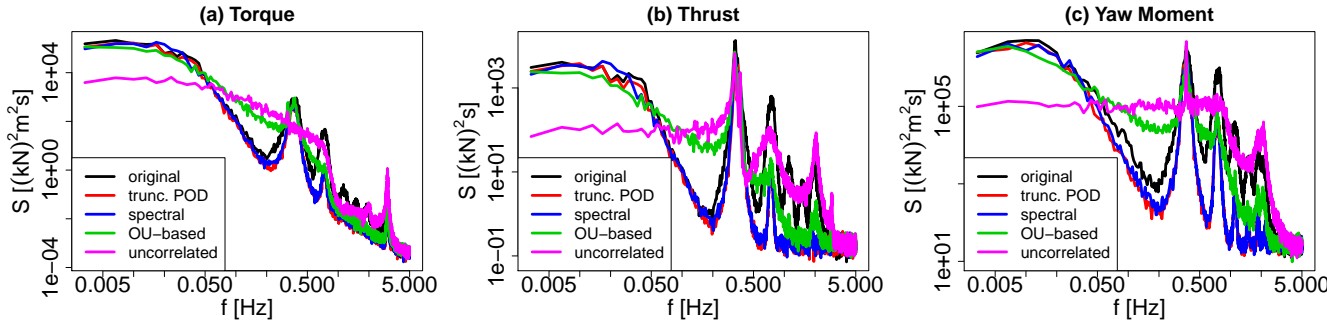

**Figure 18.** PSDs of the different loads for original LES and the different wake descriptions using $N = 6$ POD modes. Note that we aim for capturing the behavior of truncated PODs here, as pointed out in the beginning of this section.

.

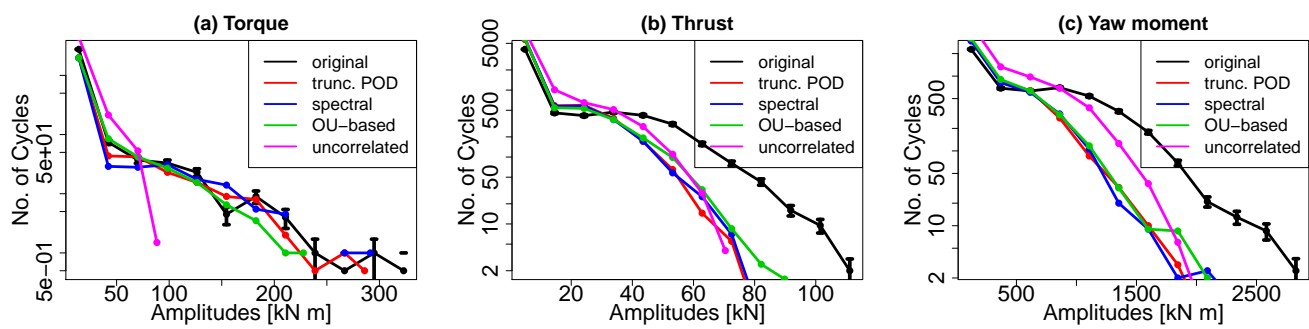

**Figure 19.** RFCs of the different loads for the original LES and the different wake descriptions using $N = 6$ POD modes. To get an impression of the estimation error, the standard error is shown for the original LES. It is estimated by $\frac{\sqrt{n_i}}{2}$ where $n_i$ is the number of half-cycles in bin $i$. Note that we aim for capturing the behavior of truncated PODs here, as pointed out in the beginning of this section.





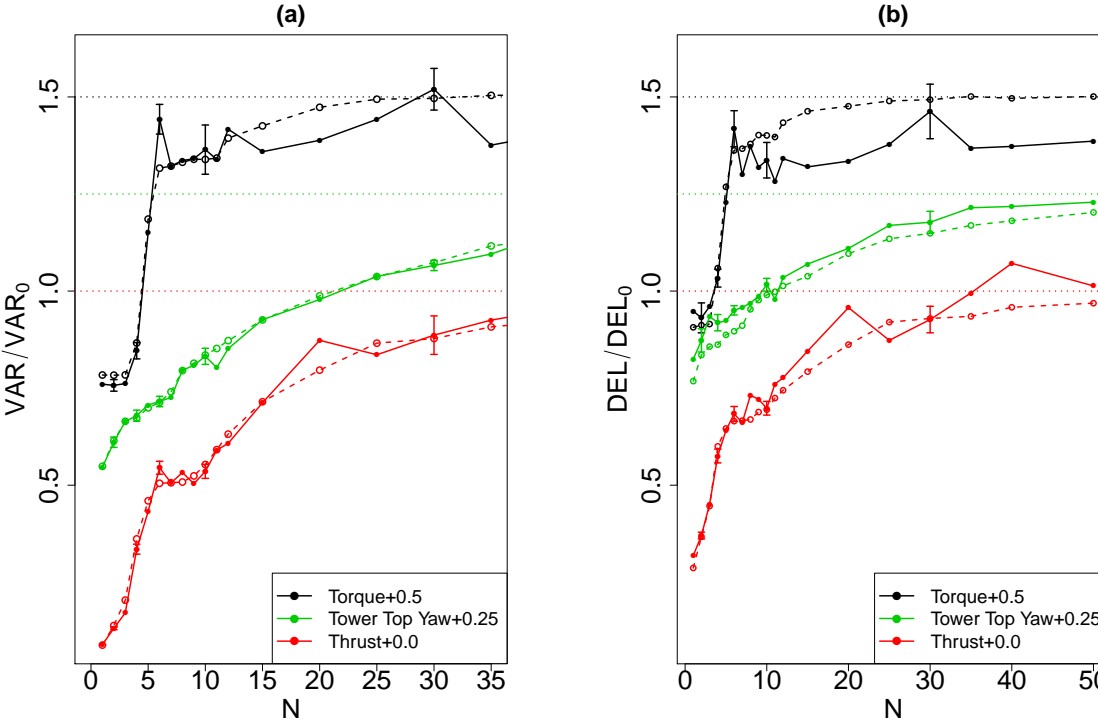

**Figure 20.** (a) Variance and (b) DELs for different loads versus no. of modes $N$ included in the wake descriptions. The solid lines with filled circles represent the results for the *spectral model* while the dashed lines show the results for the truncated POD. The curves for the different loads are vertically shifted for a better visualization, as described in the legend. The corresponding shifted values representing $1.0$ are illustrated by the dotted lines. Standard errors are shown for the *spectral model* for $N = 2, 4, 6, 10, 30$. They are estimated through the standard deviation from an ensemble of 10 realizations.

## 6   A Stochastic Wake Model with Added Turbulence

As discussed in the former sections, we need very many modes (more than 30) to capture the load behavior on small time scales and consequently also for the variance and damage equivalent loads. Including many modes leads to many model parameters which makes it difficult to build simple and practically applicable models. One way of reducing the number of model

5    parameters might be to parametrize the parameter's dependence on the mode number (e.g. Fig. 13 for the *spectral model*). However, for such a simplification, the stochastic wake models might perform weaker when taking many modes into account, as discussed in Sect. 5. Furthermore, higher order POD modes are hard to estimate from data. In this section, we examine a different approach, describing the large-scale and small-scale dynamics of the wake separately by combining the *spectral model* including a few modes with an additional homogeneous turbulent field. We start by illustrating the basic idea and underlying

10   hypothesis of our approach in Sect. 6.1. Subsequently, the performance of the extended *spectral model* with added turbulence





is investigated via the different aspects introduced in Sect. 3.5.

## 6.1 Basic Idea

As discussed in Sect. 4.2, we suspect that a few POD modes can qualitatively capture the statistical inhomogeneities of the
wake flow, as illustrated by the local TKE of truncated PODs in Fig. 7. The magnitude of the enhanced turbulence in the outer
region of the wake cannot be grasped due to missing small-scale structures in the wake deficit. Thus, here we try to capture
these structures by adding a homogeneous turbulent field to the *spectral model*. The added field is estimated directly from the
LES data, as described in the following. Inspired by the spatial structure of the local TKE, we assume that the small central
region marked by the black circle in Fig. 21a is almost always covered by the wake structure and is thus less influenced by the
large-scale motion and approximate shape dynamics of the deficit which are supposed to be described by the POD modes. We
use a three dimensional spectral surrogate of this region, as introduced in Sect. 3.4, to build a homogeneous turbulent field with
similar structures. This surrogate is shown in Fig. 21c.

The extended *spectral model* is now built as follows. The wake deficit is extracted by the same procedure as used for the
preprocessing of the field, which is described in Sect. 3.1 and illustrated in Fig. 22a-b. After changing the sign and adding
back the subtracted ABL mean field, we add the homogeneous turbulent field inside the identified wake structure. Outside the
structure, we use the atmospheric boundary layer flow from the LES which is uninfluenced by the turbine. A snapshot of the
resulting field is shown in Fig. 22c.

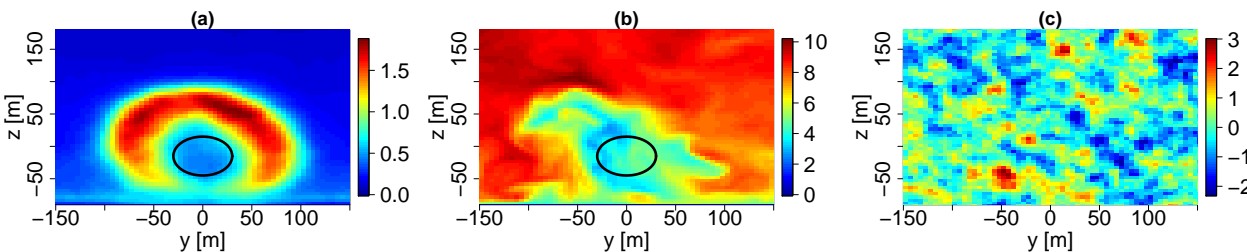

**Figure 21.** Building small-scale turbulence: (a) Local TKE $[\mathrm{m^2s^{-2}}]$ of the original LES (b) Velocity $u$ $[\mathrm{ms^{-1}}]$. The black circles mark the
central region used for generating the homogeneous turbulent field using a spectral surrogate technique (Sect. 3.4). The resulting field is
shown in (c).





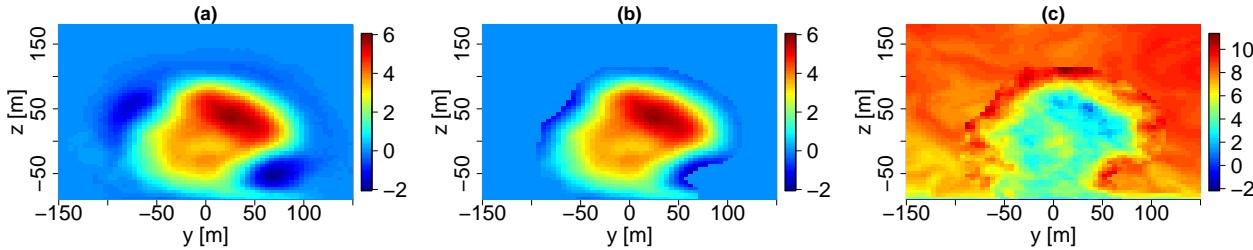

**Figure 22.** Adding Turbulence: (a) Snapshot of the *spectral model* (b) Corresponding extracted deficit structure (c) Snapshot with added small-scale turbulence in the wake and ambient ABL turbulence outside the wake structure.

### 6.2 Performance

Next, we investigate the performance of the *spectral model* combined with the additional turbulent field. A brief presentation of the results is followed by a more detailed discussion.

The local TKE of the field for different numbers of modes is shown in Fig. 23. For 6 modes, it is very similar to the original
local TKE shown in Fig. 21a, not only with respect to its spatial structure as for the truncated POD (Fig. 7) but also with respect
to its magnitude. Including much more modes leads to an overestimation of the local TKE.

The similar looking behavior of the time series of the loads for 6 modes in Fig. 24 is confirmed by the similar PSDs in
Fig. 25. Including the turbulent field leads to a good match of the PSDs for higher frequencies in contrast to the results for
the unextended *spectral model* (Fig. 18) or the truncated POD (Fig. 9). This can even be seen when using only the spectral
surrogate without any additional modes. In this case however, the low frequencies cannot be matched. With 6 modes low and
high frequencies seem to be matched pretty well.

Concerning the RFC analysis, no good modeling for thrust and tower based yaw moment could be obtained by our POD
approach (Fig. 19 and Fig. 11) up to here. After adding the turbulent field more high amplitude cycles are found in the RFCs,
as shown in Fig. 26. The model with zero modes differs strongly from the original LES for all different loads. Six modes seem
to perform best for thrust while about 20 modes are favorable for the tower base yaw moment.

For torque and thrust, variance and damage equivalent loads show relatively good agreement with the original LES when
including around $6 - 10$ modes according to Fig. 27. More modes are needed for the tower base yaw moment. Including many
more modes leads to strong overestimations for all loads, as we will discuss below.

We start the discussion of the results above by considering the local TKEs which indicate that the statistical inhomogeneities
in the field can be grasped nicely by 6 modes. Interestingly, 6 modes also seem to perform well or best for most of the other
measures such as the PSDs, RFCs, variance, or DELs of the different loads. This indicates that most relevant large-scale effects
are captured by 6 modes while the small-scale effects are captured by the homogeneous turbulent field which is moving with
the wake structure. The overestimations for higher numbers of included modes, observed particularly for the variances and
DELs or the RFCs of the thrust, are caused by the fact that higher order modes describe small-scale structures which are also



included in the added turbulent field. Thus, further adjustments of the turbulent field and the selected modes are needed. The weaker performance for the tower base yaw moment, when using 6 POD modes, appears to be caused by the fact that the truncated decompositions have problems to capture wake structures which have moved far away from the mean position of the wake, as discussed in Sect. 4.2 and Bastine et al. (2015b). It should be noted that in principle, other more sophisticated

5   homogeneous fields can be used such as found in Mann (1998) and Kleinhans (2008). However, as indicated by Bastine et al. (2015a), it might be possible that wake turbulence might be modeled similar to ideal homogeneous isotropic turbulence quite independently from the ambient flow.

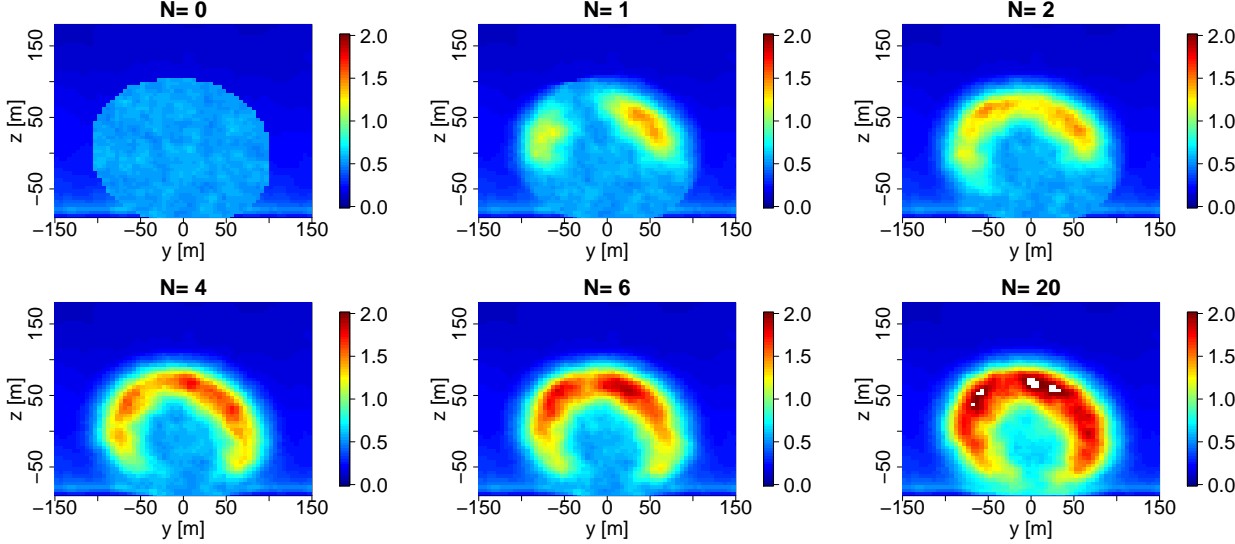

**Figure 23.** Local TKE $\langle u'(y,z)^2 \rangle_t$ [m$^2$s$^{-2}$] for original LES and the *spectral model* with added turbulence including different numbers of modes $N$.

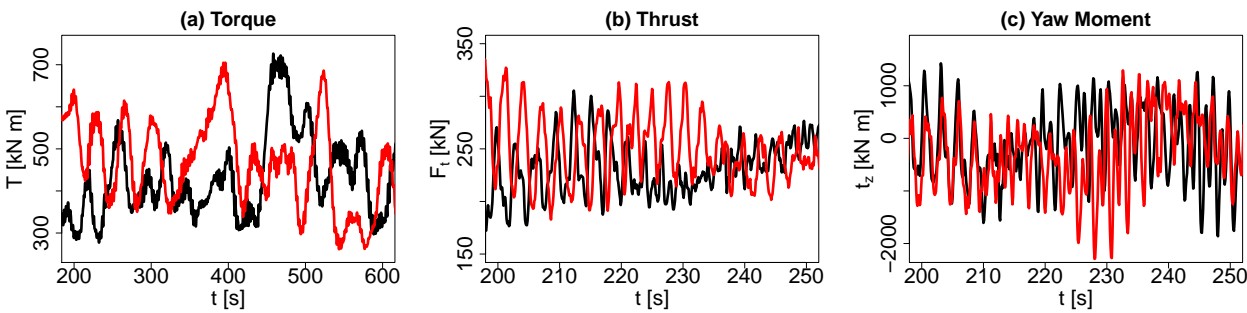

**Figure 24.** Time series of the different loads for original LES and for the *spectral model* with added turbulence including $N = 6$ modes.





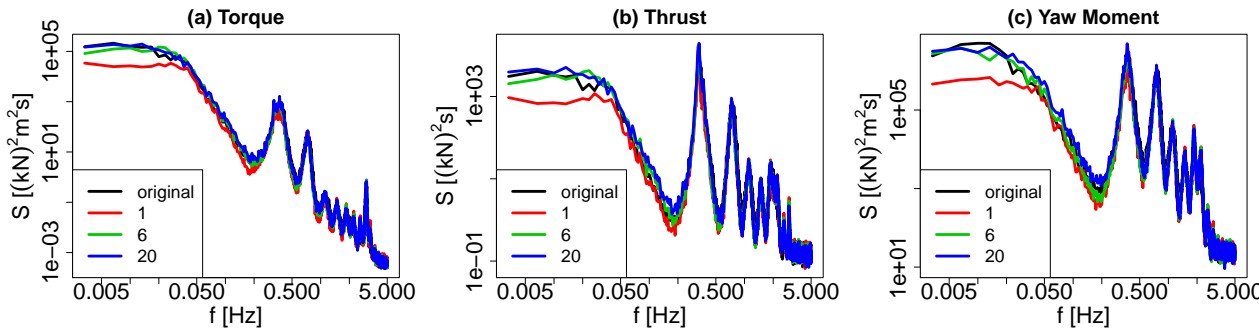

**Figure 25.** PSDs of the different loads for original LES and *spectral model* with added turbulence including different numbers of POD modes.

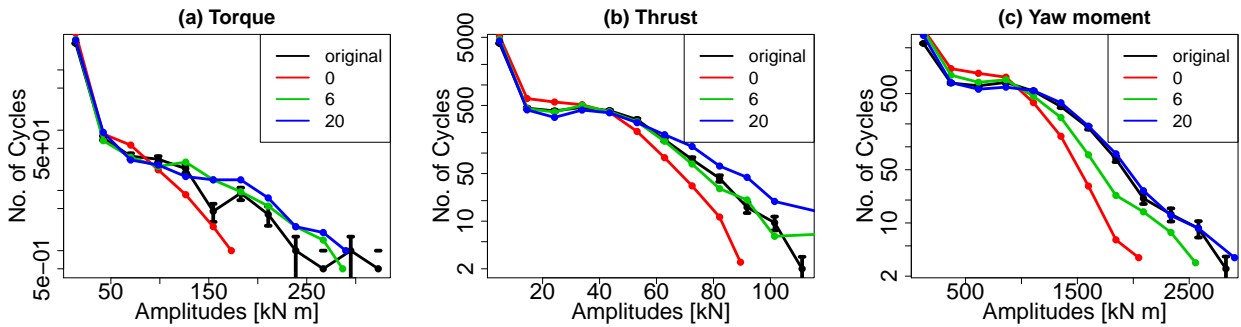

**Figure 26.** RFCs for different loads and different numbers of modes included in the *spectral model* with added turbulence.





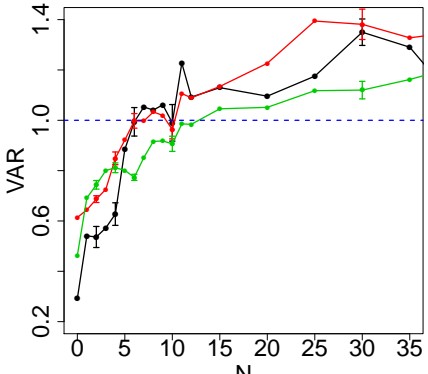 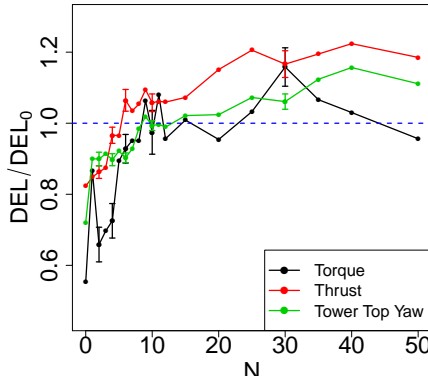

**Figure 27.** Variance and DELs for different loads versus no. of modes included in the *spectral model* with added turbulence. Standard errors are shown for the *spectral model* for $N = 2, 4, 6, 10, 30$. They are estimated from the standard deviations from an ensemble of 10 realizations.

## 7  Conclusions

In this work, is has been shown that combining the POD with stochastic weighting coefficients is a promising approach. Simple models with a few POD modes combined with simple stochastic processes could capture important aspects of the wake flow and also its impact on a wind turbine.We illustrated that the necessary complexity of the wake models strongly depends on the

quantities the models should be able to reproduce. This is not only true for the necessary number of modes, as also discussed in Bastine et al. (2015b); Saranyasoontorn (2005, 2006), but also for the complexity of the stochastic models describing the temporal dynamics through the weighting coefficients $a_j(t)$. While a trivial model for the $a_j(t)$, given by independent Gaussian numbers, is enough to grasp the spatial dependence of the average TKE, more complexity is needed to capture e.g. the temporal correlations of different loads.

Hence, our approach allows to build models which are as complex as necessary but as simple as possible with respect to a specific application. A major challenge is now to find such an optimal reduction through the selection of modes and the choice of stochastic models for the $a_j(t)$. The LES and aeroelastic simulations used in this work have proven to be a useful tool for this purpose. Further investigations using for example high-speed PIV measurements in laboratory wake flows would be very interesting. Due to their growing potential also lidar scanners could offer complementing measurements in the near future, e.g.

following ideas presented by Beck et al. (2015) and Barthelmie et al. (2016).

The results of our work also revealed that not only the large scales of the wake are relevant but that also the small-scale structures play a very important role, e.g. for capturing the RFCs of a load time series. We demonstrate that the small scales can be captured by adding a homogeneous turbulent field to our model while the large scales are captured by a few POD modes. Our approach, as well as others such as the DWM (Larsen et al., 2007a, 2008; Madsen et al., 2010), can therefore strongly

benefit from a good description of small-scale wake turbulence. Hence, its detailed investigation, as for example found in




Chamorro et al. (2012); Iungo et al. (2013); Melius et al. (2014a, b); Singh et al. (2014); Bastine et al. (2015a), is a very essential area of research for the improvement of future dynamic wake models.

In the LES used here, a quasi-steady atmosphere is simulated and therefore the large-scale dynamics of e.g. the wind direction and consequently the wake motion (meandering) are not modeled completely realistic. Thus, another possible improvement

of our model could be the inclusion of this large-scale motion through the combination with existing approaches, such as the DWM. In the DWM, the meandering caused by large atmospheric structures is the only large-scale effect which is modeled explicitly and thus a POD-based approach is a promising extension. Additionally, our model should be compared to very recent approaches by Doubrawa et al. (2016), which stochastically model the wake structure based on spectral descriptions of its edges.

It should be noted that our work is to be considered as a proof of concept illustrating a possible way of modeling dynamic wind turbine wakes. Our ideas have to be tested and probably extended for different atmospheric situations, different wind turbines or distances from the turbine. Having found useful parametrizations from high performance flow simulations or elaborated experimental investigations, our proposed procedure leads to wake flows which can be generated in a very fast stochastic manner. This enables us to perform long-term studies or large numbers of simulations which could be especially interesting

when considering wake flows in the design process of wind turbines. Furthermore, the controlling of wind turbines in wind farms could strongly benefit from real-time calculations of wake flows.

Furthermore, we would like to stress that the idea of combining modal decompositions, such as the POD, with stochastic models is not only applicable to wind turbine wakes. In principle it might be of use for the description of any complex dynamical system, as for example given by fluid flows. Similar stochastic approaches could also extend existing methods such

as the dynamic mode decomposition or dynamic models derived from the projection on the Navier-Stokes Equation. This way the impact of missing information, such as discarded weighting coefficients, could be lumped into stochastic fluctuations.

*Author contributions.* David Bastine performed the analysis and wrote the major part of this article. Joachim Peinke and Matthias Wächter are supervisor and co-supervisor of David Bastine and his current Ph.D. work. Therefore, they helped with the ideas and discussions and proofread the manuscript. Lukas Vollmer performed the LES simulations and mainly wrote the part describing the LES data.

*Competing interests.* The authors declare that they have no conflict of interest.

*Acknowledgements.* We would like to acknowledge fruitful discussions with Philip Rinn, Björn Witha, Pedro Lind, Hauke Beck, Christian Behnken, Davide Trabucchi and Juan Jose Trujillo from ForWind Oldenburg). This work has been funded by the Bundesministerium für Wirtschaft und Energie (BMWi) due to a decision of the German Bundestag (FKZ0325397A) and by the Lower Saxony Ministry of Science and Culture within the project "ventus efficiens".



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
