# Peer review of "Stochastic Wake Modeling Based on POD Analysis"

_Wind Energy Science, 2016_

## Referee Comment (RC1) · Anonymous Referee #1 · 25 Nov 2016

**General comments**

The highlight of this paper is the integration of a POD approach to model the low-frequency energy content of the wake, with a homogeneous turbulent field in the center of the wake to model the high-frequency flow fluctuations. The work is innovative and of high value to the scientific community, particularly in that it can potentially be combined with existing wake models which consider other features (e.g. large-scale meandering) that are not present in this framework. Additionally, the authors present three different methods of estimating the coefficients of the POD modes. These results lay out the framework necessary to extend the model to a wider range of atmospheric stability conditions, which is necessary in order to generalize the model so that it can be used for long-term studies and controls applications as suggested in the conclusion.

The main weakness of the paper is its length -- there is a lot of information packed into it and although the content is very interesting, it gets a little tedious after a while due to the repetitiveness of the figures and the organization of the results. A suggestion: can you give another review of the text and figures to ensure that there is no way to compact it a bit? Some figures are there but only one sentence is given about them, are they really necessary?

Finally, by the end of the paper it is still not clear what a good model performance is i.e. what are the authors aiming for with this model?

**Specific comments**

P5
L5: What are the criteria to define satisfactory performance? If you get to them later, just say here that you will get to them later. Same issue P7 L29.
P6
L2: "temporally local": can you elaborate a bit more on what this means and how it affects your analysis? ie, at each snapshot the wake-defining vd value is different? Can you say how much this value changes over time? Since it depends on the max vd, which is a very unsteady quantity, this could criterion could also oscillate a lot depending on your TI?
At this point I am confused as to why "the stochastic modeling approach, presented in the following, does not principally rely on the chosen preprocessing procedure" but I assume it will be clear as I continue reading the manuscript.
P12
L15: "The POD modes also reveal this non-symmetric behavior of the wake." Would this also be the case if you were looking at a downstream distance > 3.5 D?
L18: "As discussed in Bastine et al. (2015b), mode 1 is related to the horizontal large-scale motion of the wake." What is meant by horizontal here, its downstream advection or its cross-stream meandering? Because your cross-stream component is zero here, right? OK you briefly discuss this in P15 L7-10 but have a think about whether the controller also may be driving this.
P14

L19: why are a lot fewer modes needed to reproduce the torque? could this be just a consequence of how your variable speed controller works, ie it is driven by your hub height wind speeds and does not respond to small scale fluctuations?
P23
L25-26: and what does this mean in terms of fatigue loading / premature failure? Would you be able to add a sentence commenting on this, if not quanti- then at least qualitatively?
P26
Figure 20: why is torque so off in your stochastic models, while it is the best for the truncated POD? Also why standard errors for the spectral model only are given? In one of my comments above I wondered about the variation in the uncorrelated model, can you include that?
P31
L16: your work didn't really reveal this but just confirmed it. It is pretty well known that small-scale fluctuations are extremely relevant for fatigue loading.
Conclusion
"models which are as complex as necessary", but what is necessary for fatigue life estimation? After your discussion this question of how much we care about these high frequencies really remains unanswered?
Do you have any ambition to validate these spectra or TKE values with measurements? If so, mention it here perhaps where you have the LiDAR comment.

**Technical corrections**

* After defining an acronym, always use it instead of spelling out the acronym again several times.
* Check carefully punctuation, extra spaces, extra empty lines.
* Remove use of non-scientific terminology, e.g. "we suspect", "grasp", "lumped"

P1
L6: for->to
L11: procedure, how to -> procedure toL14-15: it still remains an open question which features of the wake flow have to be taken into account-> which features of the wake flow to take into account remains an open question.
P2
L25: PIV-Data -> PIV data
P3
L3: Calman-filter -> Kalman filter
L15-16 : not sure what you mean by words "principle" / "principally"
P4
L11: "a medium rough sea surface" word medium sounds weird here, you mean moderately rough?
Figure 1 caption: please say whether looking up or downstream; should you really put the units $[\text{ms}^{-1}]$ before (a), (b), (c) if it's not the unit for all frames? ie (c) is different unit

P5

In Figure 2 you are using ⟨u⟩ to represent your means but you did not say in the text what the angled brackets mean? Is it temporal mean over the 7050 s, ie 23500 snapshots? (OK you define this P6 L14 so maybe move the definition up or have it twice so it is given before Figure 2, or in the caption).

Figure 3 caption: please say whether looking up or downstream.

L4: LES Data -> LES data

L5: satisfying -> satisfactory

L10: similarly -> remove this word

L11: upstream the -> upstream of the

P6

L5: remove both commas; again confusing use of the word "principally"

L7: have been -> either "were" or "are"

L8: lead -> led

Figure 4: I don't understand why vd is not given in the conventional definition where it is a fractional value normally between 0 and 1?

P7

L12: can you explicitly say what indices i,j will refer to throughout manuscript

L26: lumped into -> "described by" or some other term

L29: satisfying -> satisfactory

P8

L5: repetitive and confusing sentence

P9

L5: already defined PSD so no need to spell it out again

L21-22: sentence is missing a verb?

P10

L6-7: "In this paper, we use three of the multiple loads calculated by FAST , namely the rotor torque T, the rotor thrust $F_t$ and the tower base yaw moment in z-direction $t_z$."->"In this paper, we focus on rotor torque T, rotor thrust $F_t$ and tower base yaw moment in the z-direction $t_z$."

L16: energies -> energy

L17: " since they are commonly used" -> remove this

L23: missing a period

L28:   shown in Fig. 1a. -> shown in Fig. 1a. that

You say "turbulent kinetic energy" a lot before here, so define TKE when it first appears in the manuscript

P11

L13: hy- drodynamics-> fluid dynamics or aerodynamics

L18: since this will come back in your analysis, I think you should explain here how you came up with a f_rot ~ 0.12 Hz for your average value. Or explain it in second paragraph of chapter 2 where you give rotor characteristics

P12

L4: grasp -> reproduce

L10: " There is slight tendency from larger to smaller structures with increasing mode number", confusing sentence

L12: than the modes -> than those

L18: non-axisymmetry -> axial asymmetry

L19: " the fact that we do not find a similar mode representing the motion in another direction", not very clear what you mean

L23: In the spirit of -> please reword

P13

Figure 5: add punctuation to caption

Figure 6: again, looking upstream or downstream? is this one realization of your uncorrelated model, ie how were the weighted coefficients generated for these figures?

P14

L2: yielding truncated PODs -> remove this, redundant

L24: less -> fewer

L25: fix white space between paragraphs

L27: suspect -> hypothesize?

P15

L19: fix white space between paragraphs (this happens a lot in the manuscript is this a formatting requirement?!)

L22: for -> towards

P16

Figure 7 caption: can you add (a) through (f) labels (consistently with your other figures) and reference that in your caption accordingly ie Local TKE $\langle u^{'}(y,z)^2\rangle_t$ $[m^2 s^{-2}]$ for original LES (f) and truncated PODs including different numbers of modes N (a-e). Also once again what is the direction of the x axis...?

Figure 8 and 9 (c) title Tower Base Yaw Moment, figures should be somewhat self-explanatory without reading the entire manuscript so it's important to say it's the tower base moment, if you don't want to repeat that all the time then define an acronym TB Yaw Moment

P18

Figure 11 caption: Rainflow counting histograms (RFCs) -> Rainflow counting (RFCs) histograms

P19

L21: does the best job -> reword this

L22: has been -> is

L23: three model parameters -> three parameters

P22

L1: for capturing -> to capture

L4: lead -> leads or led

L20: truncated POD -> the truncated POD

P23

L1: does the best job -> reword

L6-7: interesting that uncorrelated model looks better than OU...One wonders why

L14: less good -> reword

L23: lead -> leads

L26: perform weaker -> reword; perhaps underperform?

P24
Figure 16: I am not an expert in POD analysis but I do think you should try to appeal to a large audience and make this as clear as possible, so I must say I still don't understand whether these are a mean wake over those thousands of snapshots? Is TKE always averaged over the entire period in your paper, as one may assume from $\langle u'^2 \rangle$?
Figure 17: might want to stay away from green+red as some people can't differentiate these? Maybe the truncated POD (ref data) should be consistently black just like in previous figures you used black for the reference data set?
P25
Figure 18 caption: for capturing -> to capture
Since the uncorrelated model is random, does it produce different spectra every time you run the model, or is it robust in terms of mean spectra / mean statistics?
Figure 17-19: Tower Base Yaw Moment
Figure 18-19: this sentence is unecessary here: "Note that we aim for capturing the behavior of truncated PODs here, as pointed out in the beginning of this section."
P26
L2: very -> remove
L3: damage equivalent loads -> DELs
P27
Figure 21: is (b) a snapshot?
Figure 22: OK it is very important here that you keep the same colorbar scale in all panels (a) through (c)
P28
L4-5: To enable direct comparison between Fig 23 and Fig 21, colorbar must be the same here too (going from 0 to 2).
L7: similar looking -> reword
L13: more high -> higher
L21: perform well or best -> reword
L22-24: didn't you just say this up above?
P29
L3: have problems -> struggle
L6-7: not sure what you are suggesting here?
Figure 23: add (a) through (f) sub captions here too.
Figure 24: Why is T time series much longer?
P30
Figure 26: Modify axes limits so as not to cut your data
Figures 24-26: Tower Base...
P32
L4: realistic -> remove word
L11: probably -> remove word

---

## Referee Comment (RC2) · Anonymous Referee #2 · 2 Dec 2016

**Review of the manuscript wes-2016-38, entitled "Stochastic Wake Modeling Based on POD Analysis", by D. Bastine, L. Vollmer, M. Wachter and J. Peinke**

This manuscript deals with a POD analysis of LES data of a single wind turbine wake, which extends results from a previous paper by the authors, Bastine et al. 2015. After a classical POD analysis and truncated POD reconstruction, two stochastic models and a spectral model for predictions of unsteady flows and loads connected with wake flows are proposed. Furthermore, an empirical technique is proposed to include small-scale turbulence in the prediction of dynamic loads.

The manuscript is very lengthy. Sometimes discussions are redundant, quite trite, or providing superfluous bridges between different sections. In contrast, key points of the work, such as description of the models, fitting procedures, are not described in detail. As reported in my detail comments, several figures and paragraphs can be completely removed.

Besides, writing and presentation, I have major concerns from the scientific and technical standpoints as well, which I am going to list in the following:
1.      From Figs. 5 and 6, I guess POD analysis has not achieved a statistical convergence and several inaccurate conclusions might have been drawn. See my detail comments 16 and 17.
2.      Performance of the uncorrelated model are extremely poor. Therefore, I recommend to remove this model from the manuscript.
3.      Nonetheless, From Figs. 14, 15, 17 show that predictions obtained with the spectral and OU-based model are very poor as well. Even if mean and standard deviation of the original signal are predicted with a good accuracy, these predictions are completely out of phase. This makes me thinking that applications of these models to real flows with a varying atmospheric stability or operative conditions of the wind turbines will lead to very poor predictions.
4.      In my opinion, the method proposed to include predictions of small-scale turbulence is quite rudimental and without any theoretical background. I am concerned that these models might fail for real atmospheric flows. Actually, we have already quite robust models, such as these cited in my detail comment 5, to reproduce synthetic turbulence or, if needed, CFD tools.

Therefore, according to my comments, I cannot recommend this manuscript for publication this time. I hope that the comments reported below might be useful for the authors.

**Detail comments:**
1.      P1L5: "…load static characteristics"; if I am not mistaken, the proposed model can only predict load fluctuations, is that right? In that case please revise your abstract.
2.      P1L2 and throughout the paper: "which" typically goes after a comma.
3.      P2L32: "… differential equations can be obtained by projecting…" I guess you mean performing Galerkin projection
4.      P3L3: there is a typo, Kalman.

5.	P3L16: "it is principally possible to capture the small-scale properties of the flow by adding a homogeneous turbulent field to the wake structure modeled by the POD-based approach". In my opinion this is theoretically incorrect and, thus, it lacks of generality for the model. The model can be satisfactory from a statistical standpoint because approaching smaller and smaller scales turbulence becomes more isotropic. However, turbulence theory clearly indicates that there are specific relations between correlations and energy content at different scales, which vary for different characteristics of the specific turbulent flow. A good example to produce a synthetic turbulent signal is the Mann's model (J. Mann, The spatial structure of neutral atmospheric surface-layer turbulence, JFM, 273, 141-168, 1994), or the modified version for stably stratified flows proposed in A. Segalini et al., A spectral model for stably stratified turbulence, JFM, 781, 330-352, 2015.

6.	Fig. 1: the mean velocity field looks skewed in the vertical direction. Some comments are reported later in the paper. Please provide your justifications here.

7.	P4L21: "Snapshots of this plane are shown in Fig. 3 revealing a variety of shapes of the wake structure". This information is trite. I suggest removing text and related figure.

8.	P5:4: Revise Data in data.

9.	Fig. 4: You filter out data with deficit lower than 40% of the maximum deficit. The maximum deficit is about 4, thus any value lower than 1.6 should be removed. How is it possible you still have negative values?

10.	P7L24-30: Please rephrase this paragraph. It is quite cumbersome.

11.	Sect. 3.3: The stochastic methods are described too quickly and it is difficult to get the main differences among them. I suggest dividing this section is sub-sections for each model.

12.	P11L9: explicit to which models belong to u or $\tilde{u}$.

13.	P11L9: Remove "This discussion will enable us to gain a deeper understanding of the results presented in the next sections 4-6." That's obvious, and as it should be indeed. Please remove this sentence.

14.	P11L12: " flow structures in the rotor plane change in time due to the hydrodynamics of the flow field". What do you mean for hydrodynamics of the flow field?

15.	P11L19-28: I suggest to remove it. It is a quite obvious discussion.

16.	Fig. 5a: Can you show the convergence of the POD eigenvalues and POD modes of interest for different numbers of snapshots and different sampling time?

17.	Fig. 6. POD modes typically capture flow dynamics as couple of two POD modes with about same energy content (POD eigenvalues), spectral content, but they are orthogonal. Your first POD mode is clearly isolated and decoupled from the other modes. Therefore, it should not be associated with flow dynamics. In contrast, this might be a sign of not-achieved convergence of the POD analysis. If you try to reduce the number of snapshots, then energy of this mode should increase. Can you please verify my speculation?

18.	Fig. 6: Showing the POD modes does not provide any essential information. I would save space by removing this figure.

19.	P18L18: Explain more in detail this fitting procedure.

20.	P18L19: "$S_0$ is systematically underestimated due to the logarithmic function". Why a fitting with a log function always underestimates?

21.	P18L18-P19L4: You present 2 figures (6 panels) is 6 lines. If these plots are not crucial,

then just remove them.

22.	P19L6-L19: Since here and in the following you will show that the uncorrelated model is highly inaccurate (see Fig. 14, 15c, 17 etc.). Then, why do you present this model? In my opinion, a scientific paper should present the main information for the community in a concise way.

23.	Fig. 14. In my opinion, these models do not provide a satisfactory prediction. Are you sure it is worth to document these results?

24.	Fig. 17. "For the *OU-based-* and 10 *spectral model*, the time series resemble the loads of the truncated POD but drawing further conclusions from a single short time window is difficult", In my opinion, the model predictions are completely out of phase. Why we should learn about these models?

25.	P27L10-12: "We use a three-dimensional spectral surrogate of this region, as introduced in Sect. 3.4, to build a homogeneous turbulent field with similar structures. This surrogate is shown in Fig. 21c." This small-scale turbulence is already included in your POD modes. Why don't you try to recover this information from your POD results?

26.	P27L16: "Outside the structure, we use the atmospheric boundary layer flow from the LES which is uninfluenced by the turbine" Do you add the mean flow or the instantaneous turbulent flow? In the second case, in my opinion this procedure is theoretically incorrect. You can find a large number of papers describing interaction between wakes and boundary layer flows.

27.	Fig. 24: Is this a satisfactory prediction?

---

## Author Comment (AC1) · 13 Feb 2017

We thank the referee for the detailed revision of the manuscript. We are happy that the referee appreciates our model idea and will gladly consider the referee's comments to improve the quality of the article. In the following, we will answer the comments of the referee. The original referee's comments are written in italic letters.

**1   Answers to General Comments**

*The main weakness of the paper is its length – there is a lot of information packed into it and although the content is very interesting, it gets a little tedious after a while due*

*to the repetitiveness of the figures and the organization of the results. A suggestion: can you give another review of the text and figures to ensure that there is no way to compact it a bit? Some figures are there but only one sentence is given about them, are they really necessary?*

We see this point of the referee and agree to shorten the paper with a stronger focus on the main results. Some of the figures, which are not crucial, will be removed. Here we will take also the advise of the second referee into account.

We tried to provide in our paper all results of the considered cases. In a revised version some of our results can can be taken out of the main part of the paper and put in an appendix, if possible and accepted.

*Finally, by the end of the paper it is still not clear what a good model performance is i.e. what are the authors aiming for with this model?*

This remark of the referee shows us that we have to improve our conclusions. Actually our result is that there is is not one perfect model, but depending on the chosen aspect different simplified models are proposed.

In the end, our goal is to build stochastic models, which reproduce important aspects of the load dynamics acting on a wind turbine in the wake, such as e.g. DELs. Furthermore, these models should be as simple and efficient as possible. Due to the flexible nature of our model, different models can be taken as "good" and different levels of complexity might be needed dependent on the aspect or load of interest. Our main approach is, to use computationally expensive simulations like the highly precise LES models to deduce such appropriate stochastic models. Due to the efficiency of the stochastic models, long-term studies are made possible. We will try to make this point more clear in the revised version of the manuscript, particularly in Sect. 3.5 and in the conclusions.

As an improved beginning of our conclusions we suggest:

"In this work we presented a conceptional approach to derive stochastic reduced order wake models from costly CFD calculations such as LES simulations. Such expensive computations can be done on big computer clusters only for several hours of operation of a turbine. We show that a strong reduction of complexity is possible and important aspects of the wake flow and its impact on a wind turbine can be grasped by relatively simple models. The corresponding wake flows can be generated in a very fast stochastic manner enabling, for example, long-term studies of e.g. load assessments.

In contrast to a general model reduction of a wake flow, we show that the necessary complexity of the wake models strongly depends on the quantities the models should be able to reproduce. This is not only true for the necessary number of modes, as also discussed in Bastine et al. (2015b); Saranyasoontorn (2005, 2006), but also for the complexity of the stochastic models describing the temporal dynamics through the weighting coefficients $a_j(t)$. The value of our approach has to be judged by the application as for each new aspect or even each new turbine and possibly new simplified models have to be worked out. In our opinion such a model reduction can be done by a straight forward procedure......"

**2   Answers to specific comments**

*P5 L5: What are the criteria to define satisfactory performance? If you get to them later, just say here that you will get to them later. Same issue P7 L29.*

We propose to add "as discussed in Sect. 3.5" to P5L5. In P7L29, we removed the corresponding sentence, since it basically just repeats the same statement in P5L5.

[Figure]

*P6 L2: "temporally local": can you elaborate a bit more on what this means and how it affects your analysis? ie, at each snapshot the wake-defining vd value is different? Can you say how much this value changes over time? Since it depends on the max vd, which is a very unsteady quantity, this could criterion could also oscillate a lot depending on your TI? At this point I am confused as to why "the stochastic modeling approach, presented in the following, does not principally rely on the chosen preprocessing procedure" but I assume it will be clear as I continue reading the manuscript.*

We agree that the maximum velocity deficit is a very unsteady quantity, as shown in Fig. 1. However, what we mean by "the stochastic modeling approach, presented in the following, does not principally rely on the chosen preprocessing procedure" is explained in the following.

The goal of the preprocessing is to extract the wake structure. In other words, coherent structures which are only weakly influenced by the wake should be excluded from the analysis e.g. the light blue structures on the right of Fig. 2a. This can be achieved by many different approaches and thresholds. Our "temporally local" threshold is only one possible way. It actually turns out that using a steady threshold of $0.4$ times the maximum deficit value of all snapshots leads to similar extractions. These similar results are also caused by the additional dilation procedure, which makes the extraction relatively robust. In some sense, we just extract a relatively small part of the wake structure but keep the outer regions of the wake through dilation.

As explained in P6L5-9, we also used fixed confined spatial region for our analysis instead of a threshold, which also leads to similar results. In that case, the POD modes are slightly different and we sometimes need a bit more modes to achieve the same performance for the truncated POD as in the case with threshold extraction. However, our general procedure to obtain a stochastic reduced order model stays the same.

Additionally, we will subsitute the words "temporally local" by "for every time-step", which is one of our suggested changes in the corresponding paragraphs:

"Before the POD is applied to the data, the velocity field is preprocessed similarly as in *Bastine et al. (2015)* to focus the analysis on the wake structure. Coherent structures which are only weakly influenced by the wake should be excluded from the analysis e.g. the light blue structures on the right of Fig. 2a. The preprocessing is illustrated in Fig. 2. First, we subtract the mean field far upstream of the turbine (Fig. 1a) from the wake flow (Fig. 2a). The velocity deficit obtained after changing the sign of the field is shown in Fig. 2b. Second, we extract the deficit by using a relative threshold for every time step. This means that we set all values smaller than $40\%$ of the current deficit maximum to zero. This extraction is followed by a dilation procedure *Serra (1982)* to keep the neighboring regions which are lower than the threshold. The kernel used for the dilation is a disk with radius $20$ m. The resulting extracted deficit is shown in Fig.2c.

It should be noted that the stochastic modeling approach, presented in the following, does not principally rely on the chosen preprocessing procedure. Changing paramaters, such as the threshold value only lead to minor quantative differences. Even the analysis is performed confined to a fixed circular region around the wake center, instead of using a threshold, qualitatively similar results have been obtained. The threshold procedure is chosen to be consistent with our former work presented in *Bastine et al. (2015)* where it lead to better results concerning the selection of POD modes."

*P12 L15: "The POD modes also reveal this non-symmetric behavior of the wake." Would this also be the case if you were looking at a downstream distance > 3.5 D?*

Due to the presence of the ABL, the wake is not axisymmetric for higher distances either, as can e.g. be seen by the local TKE for a downstream distance of approx. 5 D (Fig. 3 a).

However, the wake seems to become a bit more symmetric in some sense. For

example, additionally to the mode 1 describing horizontal motion we now also obtain a mode 4 representing vertical motion ((Fig. 3b and (Fig. 3c). However, both have completely different eigenvalues (226 and 82) and thus also the large-scale motion is still not axisymmetric.

*L18: "As discussed in Bastine et al. (2015b), mode 1 is related to the horizontal large-scale motion of the wake." What is meant by horizontal here, its downstream advection or its cross-stream meandering? Because your cross-stream component is zero here, right? OK you briefly discuss this in P15 L7-10 but have a think about whether the controller also may be driving this.*

Here, we mean the cross-stream meandering in the horizontal $y-$direction. Note, that the LES simulations including the actuator disk with rotation are run with non-zero $v,w$ component. Thus the "meandering" of the wake can also be caused by $v$ and $w$ components. $v$ and $w$ are set to zero only for the approximated descriptions through truncated POD or stochastic models. Obviously the controller of the actuator disk with rotation used in the LES influences the dynamics of the wake and thus also its movement. How this happens exactly is unclear. However, this is a realistic process which we do not want to exclude in our simulations.

*P14 L19: why are a lot fewer modes needed to reproduce the torque? could this be just a consequence of how your variable speed controller works, ie it is driven by your hub height wind speeds and does not respond to small scale fluctuations?*

The first question is also discussed in P15L7-10 which we suggest to change to

"The generally simpler performance for the torque might be related to the large moment of inertia of the rotor causing the higher frequencies and the occurring spectral peaks, which are poorly captured, to play a less important role. This can be seen by
the relatively low frequency peaks in Fig. 9a and the relatively smooth time series in Fig. 8a. The more relevant low-frequency dynamics can be captured by low order POD modes corresponding to relatively large coherent structures. Hence, the time scales relevant to a specific load strongly influence the number of modes necessary for a satisfying description of this load."

As we recently found, the torque signal put out by fast is not to be understood as the instantaneous torque obtained from summing up the tangential forces over all blade elements. It actually is calculated as a function of the rotational speed and give the same output as the generator torque. It is thus a reaction to the tangential forces. Consequently, the large moment of inertia of the rotor leads to relatively slow dynamics and a smooth signal. We will therefore refer to this torque as the generator torque instead of rotor torque in the revised version of the manuscript.

Obviously, the controller also influences the dynamics. It is actually driven by the rotational speed and is based on a relatively coarse lookup table. It therefore has relatively long response time. However, we are not convinced that this results in less high frequency dynamics of the loads such as the torque.

*P23 L25-26: and what does this mean in terms of fatigue loading / premature failure? Would you be able to add a sentence commenting on this, if not quanti- then at least qualitatively?*

In the revised version of the manuscript we will change P23L25-28 to:

"It should be noted that the stochastic wake models, presented in this section, still have the same shortcomings as the truncated POD, namely the missing energy for small-scale dynamics when including only a few modes. Consequently, the fatigue of a wind turbine component will be severely underestimated if the estimation is solely

based on these low order stochastic models. Thus, in the next section we investigate an approach to include small-scale structures without the inclusion of a large number of modes."

*P26 Figure 20: why is torque so off in your stochastic models, while it is the best for the truncated POD? Also why standard errors for the spectral model only are given? In one of my comments above I wondered about the variation in the uncorrelated model, can you include that?*

This topic has been addressed P22L21 and discussed in P23L13-17. However, we will slightly rephrase P23L13-17 to:

"It is difficult to identify the reason for the possibly weaker performance for the torque when including more modes. One possible explanation might be that the parametrization of the PSD or the fitting procedure might be less good for higher mode numbers ($j > 15$). However, preliminary experiments with spectral surrogates of the $a_j$ , which match the PSD exactly, indicate similar trends. Another reason could be that the assumption of independence of the $a_j$ becomes problematic when including many modes."

The issue of the standard errors has been addressed in P22L22-25. We have slightly rephrased it to

"The errors shown are estimated as the standard deviation for an ensemble of $10$ realizations of the *spectral model*, calculated for $N = 1, 4, 5, 6, 10, 30$. It should be noted that the statistical estimates for the truncated PODs also have errors. Due to the highly turbulent nature of the flow, every LES simulation would lead to slightly different values. However, these errors are much more difficult to estimate. Running an ensemble of LES, for example, is computationally too expensive and other statistical methods lead to relatively unreliable estimates. As a first guess, we can assume that these errors are of the same order of magnitude as for the *spectral model*."

*P31 L16: your work didn't really reveal this but just confirmed it. It is pretty well known that small-scale fluctuations are extremely relevant for fatigue loading.*

We agree and will change the "revealed" to "confirmed".

*Conclusion: "models which are as complex as necessary", but what is necessary for fatigue life estimation? After your discussion this question of how much we care about these high frequencies really remains unanswered?*

The answer to this question depends on the load we aim to reproduce, as also discussed in the answer to your second general comment. It turns out that a few POD modes, e.g. $6 - 10$ for the rotor thrust, plus the inclusion of additional homogeneous turbulent field are necessary to reproduce realistic RFCs and a DEL in the right order of magnitude ($\pm 5\%$). We will slightly rephrase the corresponding paragraph starting at P31L10 to

"Hence, our approach allows to build models which are as complex as necessary but as simple as possible with respect to a specific application. Having chosen a quantity, which should be reproduced, the major challenge is to find such an optimal reduction through the selection of modes and the choice of stochastic models for the $a_j(t)$. The LES and aeroelastic simulations used in this work have proven to be a useful tool for this purpose. For example, in order to reproduce the DEL of the original LES for the rotor thrust within a few percent, approximately 6 POD modes plus a homogeneous turbulent field were necessary. Further investigations using for example high-speed PIV measurements in laboratory wake flows would be very interesting. Due to their growing

potential also lidar scanners could offer complementing measurements in the near future, e.g. following ideas presented by *Beck et al. (2015)* and *Barthelmie et al. (2016)*."

Additionally, we would like note that our main goal is to present a procedure to deduce reduced order stochastic wake models. The question what is necessary for a reliable fatigue life estimation is beyond the scope of this work since it is not even clear if DELs based on RFCs are a reliable tool for this purpose.

*Conclusion: Do you have any ambition to validate these spectra or TKE values with measurements? If so, mention it here perhaps where you have the LiDAR comment.*

Even though the main purpose of this work was a proof of concept concerning the presented stochastic modelling approach it would obviously still be interesting to validate the results the LES through lidar and load measurements. Unfortunately, there are currently no concrete plans to do so.

**3  Technical Corrections**

In the corresponding section of the referee comments, the referee makes technical comments and suggestions concerning e.g. our use of the English language. We agree with most the referee's comments and will modify the manuscript accordingly when a revised version of the manuscript is requested by the editor.
* * *
**Fig. 1.** Maximum velocity deficit versus time.

[Figure]

**Fig. 2.** Velocity deficit before (a) and after (b) extraction for t=1000 s.

[Figure]

**Fig. 3.** Results for the yz-plane 5D away from the turbine (a) local TKE (b) POD mode 1. (c)
POD mode 4.

---

## Author Comment (AC2) · 13 Feb 2017

We thank the referee for the detailed revision of the manuscript which will surely help to improve the quality of our paper. However, we believe that some of the referee's concerns might be caused by central misunderstandings, which we hope to clarify in the following. Original comments of the referee are written in italic letters. A revised version of our manuscript will be handed in when requested by the editor.

[Figure]

**1 Answers to major comments and related detail comments**

*The manuscript is very lengthy. Sometimes discussions are redundant, quite trite,or providing superfluous bridges between different sections. In contrast, key points of the work, such as description of the models,fitting procedures,are not described in detail.As reported in my detail several figures and paragraphs can be completely removed.*

We see the point of the referee that the manuscript is long but believe that relevant information is contained in our detailed results and explanations. However, the manuscript might be easier to follow when shortened with a stronger focus on the main results of the article. Further aspects we presented for completeness of our argumentations, may be put in an appendix, if this possibility is given. Thus, we will revise the manuscript accordingly taking into account the referee's comments. Furthermore, more details about the models used will be added, as also discussed in the answers to the "detail comments" of the referee.

Comments concerning the convergence of the POD
*1. From Figs. 5 and 6, I guess POD analysis has not achieved a statistical convergence and several inaccurate conclusions might have been drawn. See my detail comments 16 and 17.*
*16. Fig. 5a: Can you show the convergence of the POD eigenvalues and POD modes of interest for different numbers of snapshots and different sampling time?*
*17. Fig. 6. POD modes typically capture flow dynamics as couple of two POD modes with about same energy content (POD eigenvalues), spectral content, but they are orthogonal. Your first POD mode is clearly isolated and decoupled from the other modes. Therefore, it should not be associated with flow dynamics. In contrast, this might be a sign of not-achieved convergence of the POD analysis. If you try to reduce*

*the number of snapshots, then energy of this mode should increase. Can you please verify my speculation?*

The eigenvalues and POD modes in our work have actually converged relatively well for most values and modes of interest. For example, the first eigenvalue has reached approximate convergence already after around $1000$ s of data (see Fig. 1 left of this reply), when using a snapshot every $\Delta t = 0.6$ s. Moreover, the corresponding first POD mode stays almost the same after around $1000$ s as well (Fig. 2 of this reply). Similar results can be found for other eigenvalues and modes (Fig. 1, Fig. 2, Fig. 3 of this reply). For high mode numbers $n > 15$, the convergence of the modes sometimes gets less good, which is an additional reason for using only a few POD modes and try to capture small-scale structures in a different way, as discussed in other comments.

As requested, we also investigated the behavior of estimated eigenvalues when varying the time between the snapshots used for estimation. It turns out that when averaging over $T = 1000$ s and adding additional snapshots through reducing $\Delta t$ below $5$ s, the eigenvalues do not change strongly anymore (Fig. 4 of this reply). This indicates that a lot of redundant information is contained in such "temporally near" snapshots. However, reducing $\Delta t$ also leads to an increase of the used number of snapshots, which could also cause the observed convergent behavior. For a complete study, we would need to study convergence in the whole plane defined by $\Delta t$ and $N$ but this is beyond the scope of this work.

It should be noted again, that our work is to be understood as a proof of concept and we are sure that the relevant modes have converged well enough for this purpose. An even better convergence, which is not easy to achieve, might even improve our results. In the revised version of the manuscript, we will thus only shortly comment on the convergence of modes and eigenvalues. For example by adding

"Most of the POD modes and eigenvalues have converged relatively well when

averaging over $T > 2000$ s. However, particularly the convergence of the modes is less good for mode numbers $j > 15$. Since our work aims for a proof of concept and not for an exact estimation of POD modes and values a detailed convergence study is not presented here."

to Sect. 4.1. If requested , we could also present a convergence study in the appendix of the article.

Regarding Comment 17., we agree that the appearance of approximately degenerated (paired) eigenvalues with corresponding couples of modes does occur relatively often but by far not always. Degenerated eigenvalues often occur when symmetries are present in the flow. The corresponding eigenspace is then invariant under a certain symmetry transformation. The most important symmetry in our flow, namely the axial symmetry corresponding to rotations around the stream-wise axis, is broken due to the ABL and thus we do not expect degenerated eigenvalues.

*2. Performance of the uncorrelated model are extremely poor. Therefore, I recommend to remove this model from the manuscript.*

We agree that the results for the uncorrelated model are mostly poor except for the local turbulent kinetic energy. However, we included it in our results to present a systematic increase of complexity for the stochastic models of the weighting coefficients. In this way it can be understood how complex such models need to be, in order to capture different aspects of flow or loads on a turbine. For example, the rainflow counts of truncated POD and OU-based model are very similar. This raises the question whether an even simpler model than the OU-based model might already lead to the same results. The uncorrelated model shows that this is not the case and that at least the integral time scale of the weighting coefficients needs to be captured. These arguments will be pointed out more clearly in the revised version of the manuscript, but

at the same time we can reduce the discussion and the shown results of this simple case to a minimum.

Comments concerning the performance of the models
*3. Nonetheless, From Figs. 14, 15, 17 show that predictions obtained with the spectral and OU-based model are very poor as well. Even if mean and standard deviation of the original signal are predicted with a good accuracy, these predictions are completely out of phase. This makes me thinking that applications of these models to real flows with a varying atmospheric stability or operative conditions of the wind turbines will lead to very poor predictions.*
*23. Fig. 14. In my opinion, these models do not provide a satisfactory prediction. Are you sure it is worth to document these results?*
*24. Fig. 17. "For the OU-based- and 10 spectral model, the time series resemble the loads of the truncated POD but drawing further conclusions from a single short time window is difficult", In my opinion, the model predictions are completely out of phase. Why we should learn about these models?*

The referee's comments show that there have been some central misunderstandings. Our model Ansatz leads to a **stochastic** wake model. Thus, it only aims for matching the results of the original simulation or of truncated PODs in a statistical sense. A deterministic prediction of loads is not what our model aims for. Therefore, our predictions are obviously "out of phase". Moreover, we do not expect that a deterministic prediction of loads is possible at all. The highly turbulent nature of wake flows makes most of their behavior unpredictable due to their very sensitive dependence on small perturbations.

To avoid the aforementioned misunderstanding, we will further stress the stochastic character of our model in the revised version of the manuscript. For example, we will rephrase P10L13-14 to:

"To draw conclusions on the performance of the stochastic wake models, we compare their calculated loads with the loads for truncated PODs and the original LES. Due to the stochastic character of the models, these comparisons can only be made statistically. The load time series themselves as shown in Fig 14 (of the manuscript) can only give a visual impression on the dynamical behavior in principle."

Additionally, we consider to completely remove the figures showing the load time series. Even though they can give a first impression whether the statistical behavior looks similar, they do not offer any "real" statistical insight. Since both referees would like us to shorten the manuscript, this might be on possible way to do so.

Based on this discussion above, we do not agree that our models show poor results.

Obviously, we do not provide a complete model ready to be applied in the wind energy industry. So far, several works have discussed the POD and resulting modes as a tool for obtaining reduced order wake models (P2L19-P3L5). We simply see our work as a suggested procedure for modeling the temporal evolution of a POD-based decomposition through stochastic models for the weighting coefficients. We show that this is a promising approach since several aspects of corresponding truncated PODs and resulting loads, such as the behavior on large temporal scales, can be captured. To obtain variance and damage equivalent loads similar to the original simulation we additionally have to include small-scale turbulence. This is a further step also discussed in our article and in the answers below.

Comments concerning small-scale turbulence
*4. In my opinion, the method proposed to include predictions of small-scale turbulence is quite rudimental and without any theoretical background. I am concerned that these models might fail for real atmospheric flows. Actually, we have already quite robust models, such as these cited in my detail comment 5, to reproduce synthetic turbulence or, if needed, CFD tools.*

*5. P3L16: "it is principally possible to capture the small-scale properties of the flow by adding a homogeneous turbulent field to the wake structure modeled by the POD-based approach". In my opinion this is theoretically incorrect and, thus, it lacks of generality for the model. The model can be satisfactory from a statistical standpoint because approaching smaller and smaller scales turbulence becomes more isotropic. However, turbulence theory clearly indicates that there are specific relations between correlations and energy content at different scales, which vary for different characteristics of the specific turbulent flow. A good example to produce a synthetic turbulent signal is the Mann?s model (J. Mann, The spatial structure of neutral atmospheric surface-layer turbulence, JFM, 273, 141-168, 1994), or the modified version for stably stratified flows proposed in A. Segalini et al., A spectral model for stably stratified turbulence, JFM, 781, 330-352, 2015.*

Yes, we agree that our approach is quite simple and rudimental. It is an empirical approach and not explicitly derived from physical theory. However, in our opinion our results show that this approach leads to promising results showing that statistical inhomogeneities and large-scale dynamics can be described by a few POD modes, while a statistically homogeneous field can capture the small-scale structures. Interestingly, in a multiple wake in the laboratory, *Hamilton et al. (2016)* also find indications that discarding higher order modes "is equivalent to excluding energy homogeneously from the wake".

It should also be noted that such an empirical approach is not unusual. An established model like the DWM *Madsen et al. (2010)* also uses an empirical approach to include wake added turbulence. In their scenario a homogeneous "Mann field" with spatially dependent multiplication factor is added in the meandering frame of reference, which is also clearly an empirical approach.

In our work, we only aim for showing that the separate treatment of inhomogeneous large-scale behavior and small-scale turbulence, e.g. described by a homogeneous spectral model, is a promising approach. We chose to use a spectral surrogate of the

small- scale wake turbulence from our LES since this allows a direct comparison with the original LES simulation. Otherwise, we would also compare the ability of other models to reproduce LES turbulence which is not our goal here and beyond the scope of this work.

The sentence of the referee

*However, turbulence theory clearly indicates that there are specific relations between correlations and energy content at different scales, which vary for different characteristics of the specific turbulent flow.*

is obviously true. Our surrogate partially captures these correlations for the $u$-component since it keeps the spatial PSD of the LES, as described in Sec. 3.4. We do **not** argue that this is the best way to obtain small-scale wake turbulence models. Finding the best model for this purpose is a next step and beyond the scope of this work and should be based on existing approaches, as also proposed by the referee. A homogeneous Mann field could also be used, as already mentioned e.g. in P29L5. However, the Mann field in its original version is designed for neutral ABL turbulence and not for wake turbulence. Finding the specific properties of wake turbulence and capturing them with models like the "Mann approach" is also still a matter of current research.

Based on this discussion, we will try to make these arguments more clear in a revised version of the manuscript. For example, we will add

"The added homogeneous field is estimated directly from the LES data in the wake center, as described in the next paragraph. In this way the resulting wake model can be compared most conveniently to the original LES simulation In principle, other models such as *Mann (1998)* could be used to model the homogeneous field but in this case we would also investigate the ability of such approaches to reproduce LES turbulence which beyond the scope of this work."

to Sect. 6.1.

**2 Further Detail Comments and Answers**

*1. P1L5: "...load static characteristics"; if I am not mistaken, the proposed model can only predict load fluctuations, is that right? In that case please revise your abstract.*

You just misread. In P1L5 is written statistic characteristics, which should answer your comment. We changed "statistic" to "statistical" to avoid this possible confusion.

*2. P1L2 and throughout the paper: "which" typically goes after a comma.*

We cannot find the word "which" in P1L2. Furthermore, it is oversimplified to state that "which" typically goes after a comma. In a lot of cases a comma is not allowed before "which". For example, there must be no comma before a defining relative clause. However, we checked all our "which" sentences and found some phrases where a comma is missing. These commas will be added in the revised version of the manuscript.

*3. "... differential equations can be obtained by projecting..." I guess you mean performing Galerkin projection.*

Yes that is what we mean. For completeness, we will add
" "..., which is called a Galerkin projection." "
to the revised version of the manuscript.

*4. P3L3: there is a typo, Kalman. 5.*

We corrected this.

*6. Fig. 1: the mean velocity field looks skewed in the vertical direction. Some comments are reported later in the paper. Please provide your justifications here.*

We do not understand what you mean exactly. In the vertical direction we do not see any skewness which is not simply related to the mean field of the ABL. Which justifications do you mean?

*7. P4L21: "Snapshots of this plane are shown in Fig. 3 revealing a variety of shapes of the wake structure". This information is trite. I suggest removing text and related figure.*

We do not think that this information is trite for people not dealing with wake data from LES simulations on a regular basis. We will rephrase the corresponding sentence to

"Snapshots of this plane are shown in Fig. 3. These snapshots nicely illustrate different shapes of the wake structure, which are likely to play an important role for the loads acting on a wind turbine in the wake. As mentioned in the introduction, this is one of the major motivations for investigating a POD-based modeling approach, which can roughly describe different shapes of the wake."

*8. P5:4: Revise Data in data.*
Done.

*9. Fig. 4: You filter out data with deficit lower than $40$ maximum deficit is about $4$, thus any value lower than $1.6$ should be removed. How is it possible you still have negative values?*

Thanks you for this comment. There is sort of a typo in P6L3. It should read "This extraction is followed by a dilation procedure to keep the neighboring regions which are lower than the threshold". Maybe this already answers your question. Dilation is a standard method from image processing (see e.g. *Serra (1982)*). For completeness, we revised this part in the manuscript to:

"This extraction is followed by a dilation procedure *Serra (1982)* to keep the neighboring regions which are lower than the threshold. The kernel used for the dilation is a disk with radius $20$ m."

It turned out that without dilation we miss some outer regions of the wake structure leading to slightly too small wake structures in the truncated PODs. However, the dilation only leads to a quantitative improvement of some of our results. Qualitatively, the results with and without dilation are very similar.

*10. P7L24-30: Please rephrase this paragraph. It is quite cumbersome.*

We rephrased this paragraph to:

"Since turbulent flows such as wind turbine wakes show only statistically reproducible results, our Ansatz is to describe these $N$ weighting coefficients as a stochastic system. In this article, we additionally assume that the weighting coefficients are statistically independent yielding a description of $(a_j(t))_{j=1}^N$ by $N$ one-dimensional stochastic processes. It should be noted, that even though the assumption of independence is inspired by Eq. (6), it obviously leads to a significant approximation since the nonlinear coupling of different scales in the fluid dynamical equations is neglected. It therefore has to be justified by a satisfactory performance of the deduced model."

*11. Sect. 3.3: The stochastic methods are described too quickly and it is difficult to get the main differences among them. I suggest dividing this section is sub-sections for each model.*

We tried to keep this part short, since these are very simple standard stochastic models whose properties can be found everywhere in the stochastic literature. It is a good suggestion, however, to make the main differences between the models more clear. Thus, we will revise this section according to the referee's comment.

*12. P11L9: explicit to which models belong to u or $\tilde{u}$.*
We are not sure what you mean but we hope to clarify this by changing the corresponding sentence to
"... described by a modal decomposition, such as the truncated PODs $u^{(N)}(y, z, t)$ from Eq. (4) or corresponding stochastic wake models $\tilde{u}^{(N)}(y, z, t)$ from Eq. (7)."

*13. P11L9: Remove "This discussion will enable us to gain a deeper understanding of the results presented in the next sections 4-6." That's obvious, and as it should be indeed. Please remove this sentence.*

The sentence will be removed.

*14. P11L12: " flow structures in the rotor plane change in time due to the hydrodynamics of the flow field". What do you mean for hydrodynamics of the flow field?*

We rephrased this to simply "dynamics of the flow field", which should make things clearer. With "hydrodynamics", we just meant the dynamics of the flow field, which follow the laws of fluid dynamics.

*15. P11L19-28: I suggest to remove it. It is a quite obvious discussion.*

Thank you for this comment. We will consider removing this part in the revised and shortened version of the manuscript. It might be even possible to remove the section and move its decisive parts to the discussions of the results.

*18. Fig. 6: Showing the POD modes does not provide any essential information. I would save space by removing this figure.* We still believe that for a POD-based model some of the POD modes used should be shown. However, we will remove three modes to save space and can also remove them all if requested by the editor.

*19. P18L18: Explain more in detail this fitting procedure.*

The explanation of the fitting procedure will be changed to

"For the *spectral model*, the PSDs of the $a_j$ have to be estimated. They show a qualitatively similar behavior for all $j$ starting with a flat region for low frequencies followed by an approximate power law behavior (Fig. 12a). This form motivates the parametrization of the PSDs given by Eq. (14). The parameters $S_0$, $\alpha$ and $f_{\frac{1}{2}}$ are estimated using least squares in a logarithmic framework. This means they are obtained by minimizing $\sum_{i=1}^{N}(\log(S_i) - \log(S(f_i; S_0, \alpha, f_{\frac{1}{2}})))^2$ with respect to $S_0, f_{\frac{1}{2}}, \alpha$, where $S_i$ and $f_i$ are the PSD and frequency values obtained through the statistical estimation from the LES data. While this procedure yields satisfying estimates of $\alpha$ and $f_{\frac{1}{2}}$, $S_0$ is systematically underestimated due to the nonlinear weighting by the logarithmic function. We circumvent this problem by choosing $S_0$ to be the value which yields the estimated variance of the $a_j(t)$: $\mathsf{VAR}[\tilde{a}_j(t; S_0, \alpha, f_{\frac{1}{2}})] = \langle a_j(t)^2 \rangle_t$, where $\alpha, f_{\frac{1}{2}}$ are taken from the logarithmic fit. An example fit is shown in Fig. 12b. It should be noted that alternative fitting procedures are also possible and lead to similar results as long as the PSD is matched well in all the frequency ranges and not only for low or

high frequencies."

Fitting in a logarithmic framework is a very common procedure to find power law exponents or damping coefficients in exponential functions. However, the logarithmic function introduces a nonlinear weighting to the least squares fit. Therefore, large values of the $S_i$ have less influence than for a fit without $\log$. One resulting problem is for example that when fitting to a constant function or constant region of a function, the fit will lead to an underestimation in this region, since positive deviations are weighted weaker than negative ones. That is why we estimated $S_0$ separately.

*20. P18L19: "$S_0$ is systematically underestimated due to the logarithmic function". Why a fitting with a log function always underestimates?*

See end of former answer.

*21. P18L18-P19L4: You present 2 figures (6 panels) is 6 lines. If these plots are not crucial, then just remove them.*

It is true that we did not say that much about these figures and added them mainly for the sake of completeness. We will therefore consider removing some of them in the revised version of the manuscript.

*22. Since here and in the following you will show that the uncorrelated model is highly inaccurate (see Fig. 14, 15c, 17 etc.). Then, why do you present this model? In my opinion, a scientific paper should present the main information for the community in a concise way.*

This question has been answered in the first section when answering the major comment (2.).

*25. P27L10-12: "We use a three-dimensional spectral surrogate of this region, as introduced in Sect. 3.4, to build a homogeneous turbulent field with similar structures. This surrogate is shown in Fig. 21c." This small-scale turbulence is already included in your POD modes. Why don't you try to recover this information from your POD results?*

This is an important point. As already discussed in P26L1-9, this has multiple reasons. The most obvious one is that many modes lead to many parameters for the stochastic processes. Our idea and hope is that a homogeneous spectral model to capture the small-scale structures can be parametrized in a more simple manner. Models such as the "Mann Model" for free neutral ABL turbulence give hope that such simple models for small-scale wake turbulence can be obtained as well.

*26. P27L16: "Outside the structure, we use the atmospheric boundary layer flow from the LES which is uninfluenced by the turbine" Do you add the mean flow or the instantaneous turbulent flow? In the second case, in my opinion this procedure is theoretically incorrect. You can find a large number of papers describing interaction between wakes and boundary layer flows.*

We add the instantaneous turbulent flow. We agree that this procedure is a strong approximation and that close to the wake structure the ambient turbulence will not be statistically identical to the "wake-free" case. However, it is still possible and supported by our results that such a strong approximation can lead to a useful wake description, i.e. that this interaction region is not relevant for all aspects of loads acting on a turbine in the wake. Hence, we will add a sentence concerning the neglection of interactions to the revised of the manuscript.

*27. Fig. 24: Is this a satisfactory prediction?* This question has been answered in the former section when answering the Major comment (3.).

[Figure]

[Figure]

**Fig. 1.** Convergence of the eigenvalues: Eigenvalues estimated using temporal averaging over different times T. Here, the time between two snapshots used is 0.6 s.

[Figure]

**Fig. 2.** Convergence of POD modes: Mode 1 estimated using temporal averaging over different times T. Here, the time between two snapshots used is 0.6 s.

[Figure]

**Fig. 3.** Convergence of POD modes: Mode 6 estimated using temporal averaging over different times T. Here, the time between two snapshots used is 0.6 s.

[Figure]

**Fig. 4.** Estimated eigenvalues dependent on the time difference between two snapshots used for the estimation. Snapshots from a time window of width T=1000 s are used.